# Improving Unsupervised Constituency Parsing via Maximizing Semantic Information

**Junjie Chen[1], Xiangheng He[2], Yusuke Miyao[1], Danushka Bollegala [3]**
Department of Computer Science, the University of Tokyo[1]
GLAM – Group on Language, Audio, & Music, Imperial College London[2]
Department of Computer Science, the University of Liverpool[3]
`christopher@is.s.u-tokyo.ac.jp, x.he20@imperial.ac.uk`
`yusuke@is.s.u-tokyo.ac.jp, danushka@liverpool.ac.uk`

## Abstract

Unsupervised constituency parsers organize phrases within a sentence into a tree-shaped syntactic constituent structure that reflects the organization of sentence semantics. However, the traditional objective of maximizing sentence log-likelihood (LL) does not explicitly account for the close relationship between the constituent structure and the semantics, resulting in a weak correlation between LL values and parsing accuracy. In this paper, we introduce a novel objective that trains parsers by maximizing SemInfo, the semantic information encoded in constituent structures. We introduce a bag-of-substrings model to represent the semantics and estimate the SemInfo value using the probability-weighted information metric. We apply the SemInfo maximization objective to training Probabilistic Context-Free Grammar (PCFG) parsers and develop a Tree Conditional Random Field (TreeCRF)-based model to facilitate the training. Experiments show that SemInfo correlates more strongly with parsing accuracy than LL, establishing SemInfo as a better unsupervised parsing objective. As a result, our algorithm significantly improves parsing accuracy by an average of 7.85 sentence-F1 scores across five PCFG variants and in four languages, achieving state-of-the-art level results in three of the four languages.

## 1 Introduction

Unsupervised constituency parsing is a syntactic task of organizing phrases of a sentence into a tree-shaped constituent structure without relying on linguistic annotations (Klein & Manning, 2002). The constituent structure is a fundamental tool in analyzing sentence semantics (i.e., the meaning) (Carnie, 2007; Steedman, 2000). It can significantly improve performance for downstream Natural Language Processing systems, such as natural language inference (He et al., 2020), machine translation (Xie & Xing, 2017) and semantic role labeling (Chen et al., 2022) systems. It guides the progressive construction of the sentence semantics, as illustrated in Figure 1. Each constituent in the structure corresponds to a meaningful substring, forming partial representations of the sentence semantics. One can easily recover the full sentence semantics by gradually constructing the semantic representation of those constituent substrings. Following the observation, we hypothesize that *constituent substrings in the sentence carry significant semantic information*.

Maximizing sentence log-likelihood has traditionally been the primary training objective for training unsupervised constituency parsers (Eisner, 2016; Kim et al., 2019a). However, the Log-Likelihood (LL) function does not explicitly factor in the syntax-semantics alignment. This leads to a poor correlation between the LL value and the parsing accuracy. We will further discuss this poor correlation in Section 5.3. As pointed out in previous research, it is challenging to train a Probabilistic Context-Free Grammar (PCFG) parser that outperforms trivial baselines with the LL maximization objective (Carroll & Charniak, 1992; Kim et al., 2019a). Successful training commonly involves altering the LL maximization objective, such as imposing sparsity constraints (Cohen et al., 2008; Johnson et al., 2007) or heuristically estimating the LL value (Spitkovsky et al., 2010). Theses evidence suggests that the LL function might not provide robust information to distinguish between constituents and non-constituents, rendering LL an insufficient objective function for unsupervised parsing.

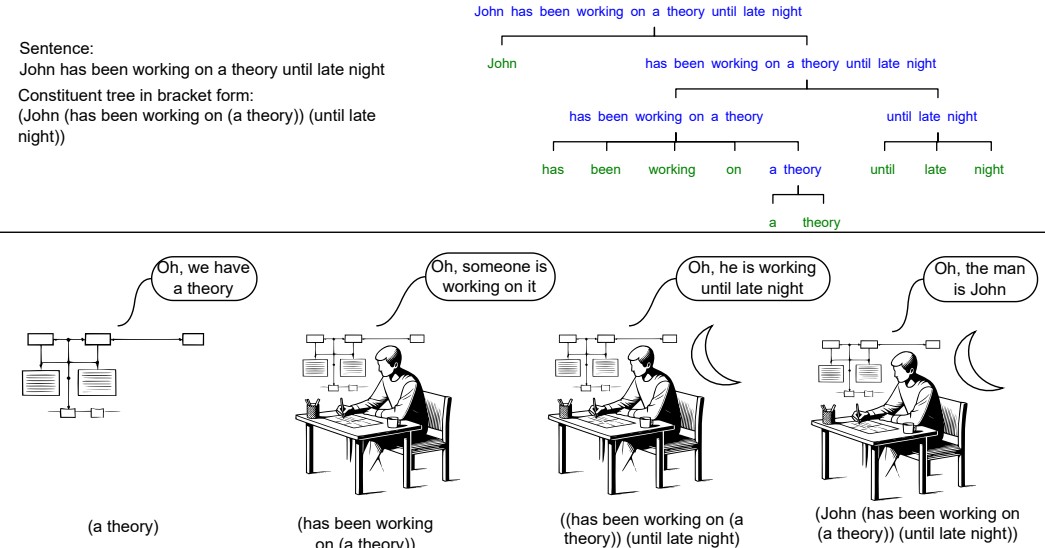

Figure 1: An illustration of the progressive semantics build-up in accordance with the constituent structure. The tree structure in the top-right shows the simplified constituent structure for illustration purposes. Constituent substrings are highlighted in blue.

In this paper, we propose a novel objective for training unsupervised parsers: maximizing SemInfo (the semantic information encoded in constituent structures). Specifically, we introduce a bag-of-substrings model to represent the sentence semantics with substring statistics, in parallel to how bag-of-words models represent document topics with word statistics. Next, we estimate the semantic information encoded in substrings (i.e., substring-semantic information) by applying the Probability-Weighted Information (PWI) metric (Aizawa, 2003) developed for the bag-of-words model to our bag-of-substrings model. Finally, we calculate the SemInfo value of a constituent structure by summing up the substring-semantic information associated with the structure. Experiments show a much stronger correlation between SemInfo and parsing accuracy than the correlation between LL and parsing accuracy. The improved correlation suggests SemInfo is an effective objective function for unsupervised constituency parsing. In addition, we develop a Tree Conditional Random Field (TreeCRF)-based model to apply the mean-field SemInfo maximization training to PCFG parsers (the state-of-the-art non-ensemble method for unsupervised constituency parsing (Liu et al., 2023)). Experiments demonstrate that the SemInfo maximization objective improves the PCFG's parsing accuracy by 7.85 sentence-F1 scores across five latest PCFG variants and in four languages.

Our main contributions are: (1) Proposing a novel method for estimating SemInfo, the semantic information encoded in constituent structures. (2) Demonstrating a strong correlation between SemInfo values and parsing accuracy. (3) Developing a TreeCRF model to apply mean-field SemInfo maximization training to PCFG parsers, significantly improving parsing accuracy and achieving state-of-the-art level results as non-ensemble parsers.

## 2 BACKGROUND

The idea that constituent structures reflect the organization of sentence semantics is central to modern linguistic studies (Steedman, 2000; Pollard & Sag, 1987). A constituent is a substring $s$ in a sentence $x$ that can function independently (Carnie, 2007) and carries self-contained meanings (Heim & Kratzer, 1998). A collection of constituents forms a tree-shaped structure $t$, which we can represent as a collection of its constituent substrings $t = \{s_1, s_2, ...\}$. For example, the constituent structure in the top right of Figure 1 can be represented as {"a theory", "until late night",...}. Previous research (Shen et al., 2017; Yang et al., 2021b) measures the accuracy of the parsing prediction by instance level sentence-F1 ($SF1^i$) score. Aggregating the $SF1^i$ score over the corpus gives the corpus-level sentence-F1 score ($SF1^c$), which previous research used to evaluate the parser quality.

In this paper, we will apply the Probability-Weighted Information (PWI) (Aizawa, 2003) designed to measure word-topic information in bag-of-words models (Figure 2a) to measuring substring-semantic information in our bag-of-substrings model. PWI is an information-theoretic interpretation of the term frequency-inverse document frequency (tf-idf) statistic. The tf-idf statistic is an effective feature in finding keywords in documents (Li et al., 2007) or in locating documents based on the given keyword (Mishra & Vishwakarma, 2015). Let $\mathcal{D}$ denote a document corpus, $d_i$ the $i$-th document in the corpus, and $w_{ij}$ the $j$-th word in $d_i$. The bag-of-words model represents the document $d_i$ as an unordered collection of words occurring in the document (i.e., $d_i = \{w_{i1}, w_{i2}, ...\}$). Tf-idf, as shown in Equation 1, is the product of the term frequency $F(w_{ij}, d_i)$ (i.e. the frequency of $w_{ij}$ occurring in $d_i$) and the inverse document frequency (i.e. the inverse log-frequency of documents containing $w_{ij}$). PWI interprets the term frequency as the word generation probability and the inverse document frequency as the piecewise word-document information (Equation 2). The PWI value estimates the information that $w_{ij}$ carries with regard to $d_i$. A high value indicates that $w_{ij}$ is both frequent in $d_i$ and strongly associated with $d_i$. In other words, $w_{ij}$ is a keyword of $d_i$.

$$\text{tf-idf}(w_{ij}, d_i) = \underbrace{F(w_{ij}, d_i)}_{\text{term frequency}} \times \underbrace{\log \frac{|\mathcal{D}|}{|d' : d' \in \mathcal{D} \wedge w_{ij} \in d'|}}_{\text{inverse document frequency}} \tag{1}$$

$$\approx \underbrace{P(w_{ij}|d_i)}_{\text{word generation probability}} \times \underbrace{\log \frac{P(d_i|w_{ij})}{P(d_i)}}_{\text{piecewise word-document information}} \tag{2}$$

$$= PWI(w_{ij}, d_i)$$

Our method is developed upon the finding of Chen et al. (2024): constituent structures can be predicted by searching for frequent substrings among semantically similar paraphrases. We extend their findings, interpreting the substring frequency statistic as a dominating term in our proposed substring-semantics information metric and applying it to improve unsupervised PCFG training. As we will see in Section 5.2, our method significantly outperforms theirs in three out of the four languages tested.

PCFG is currently the state-of-the-art non-ensemble model for unsupervised constituency parsing (Liu et al., 2023; Yang et al., 2021a). Previous research trains binary PCFG parsers on a text corpus by maximizing the average LL of the corpus. PCFG is a generative model defined by a tuple $(NT, T, R, S, \pi)$, where $NT$ is the set of non-terminal symbols, $T$ is the set of terminal symbols, $R$ is the set of production rules, $S$ is the start symbol, and $\pi$ is the probability distribution over the rules. The generation process starts with the start symbol $S$ and iteratively applies non-terminal expansion rules ($A \to BC : A, B, C \in NT$) or terminal rewriting rules ($A \to w : A \in NT, w \in T$) until it produces a complete sentence $x$. We can represent the generation process with a tree-shaped structure $t$. The PCFG assigns a probability for each distinct way of generating $x$, defining a distribution $P(x, t)$. The Inside-Outside algorithm (Baker, 1979) provides an efficient solution for computing the total sentence probability $P(x) = \sum_t P(x, t)$. It constructs a $\beta(s, A)$ table that records the total probability of generating a substring $s$ of $x$ from the non-terminal $A$. The sentence probability can be calculated as $P(x) = \beta(x, S)$, the probability of $x$ being generated from the start symbol $S$. The $\beta(x, S)$ quantity is commonly referred to as $Z(X)$ (Eisner, 2016). Besides the total sentence probability, the $\beta$ table can also be used to calculate the span-posterior probability of $s$ being a constituent (Eisner, 2016) (Equation 3).[1]

$$P(s \text{ is a constituent}|x) = \sum_{A \in NT} \frac{\partial \log Z(x)}{\partial \log \beta(s, A)} \tag{3}$$

Span-based TreeCRF model is widely adopted in constituency parsers (Kim et al., 2019b; Stern et al., 2017). It models the parser distribution $P(t|x)$, the probability of constituent structure $t$ given $x$. It determines the probability of $t$ by evaluating whether all substrings involved in the structure are constituents. It assigns a high score to a substring $s$ in its potential function $\phi(s, x)$ if $s$ is likely a constituent and a low score if $s$ is unlikely a constituent. Subsequently, It can represent the parser distribution as $P(t|x) \propto \prod_{s \in t} \phi(s, x)$. In previous research, $\phi(s, x)$ has been parameterized differently, such as using the span posterior probability for decoding ($\phi(s, x) = P(s \text{ is a constituent}|x)$) (Yang et al., 2021b) or using the exponentiated output from Long-Short Term Memory model ($\phi(s, x) = \exp(LSTM(x, s))$) (Kim et al., 2019b).

---

[1]We explain the derivation in more detail in Section A.2.

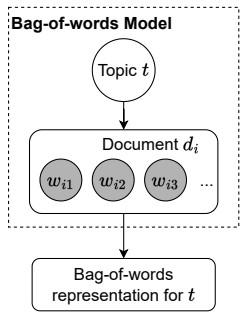

(a) Bag-of-words model.

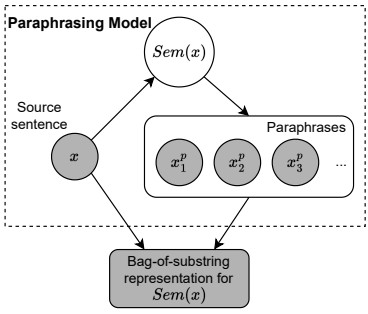

(b) Bag-of-Substrings model.

Figure 2: Parallel structure between the traditional bag-of-words representation of topics and the proposed bag-of-substrings representation of semantics.

# 3 SEMINFO: A METRIC OF SEMANTIC INFORMATION ENCODED IN CONSTITUENT STRUCTURES

In this section, we introduce our estimation method of SemInfo, the semantic information encoded in constituent structures. We first propose a bag-of-substrings model (Figure 2b), representing the semantics of a sentence by examining how substrings in the sentence are *regenerated* during a paraphrasing process. We assume the paraphrasing process is capable of generating *natural language paraphrases* (i.e., the paraphrases should both be acceptable as natural language sentences and have similar semantics to the original sentence). We use instruction-following large language models (LLMs) for the paraphrasing model, exploiting their outstanding zero-shot learning capability (Chia et al., 2023). Next, we apply the PWI metric (Aizawa, 2003) to measure the substring-semantics information, utilizing the parallel structure between the bag-of-words model and our bag-of-substrings model (Figure 2). Finally, we estimate the SemInfo value for constituent structures by summing the substring-semantics information associated with the structure.

## 3.1 DEFINING SUBSTRING-SEMANTIC INFORMATION USING BAG-OF-SUBSTRINGS MODEL

Our bag-of-substrings model shares a parallel structure with the traditional bag-of-words model. As discussed in Section 2, the bag-of-words model can model the word-topic information using the PWI metric. Exploiting the structural parallelism, we can apply the PWI metric to our bag-of-substrings model to estimate the information between substrings and sentence semantics (Equation 4).

The bag-of-substrings model is based on the paraphrasing model $P(x^p|Sem(x))$ shown in Figure 2b. The paraphrasing model takes a source sentence $x$ as input, internally analyzes its semantics $Sem(x)$, and generates a paraphrase $x^p$. We can repeatedly sample from the process, collecting a paraphrase set $\mathbb{X}^p = \{x_1^p, x_2^p, ...\}$. We define the bag-of-substrings model by examining whether a substring $s$ of $x$ appears in $\mathbb{X}^p$. We consider the appearance of $s$ in $\mathbb{X}^p$ as $s$ being generated by the bag-of-substrings model. The generation modeling establishes a relationship between the semantics $Sem(x)$ and the substring $s$, which we will use to estimate the substring-semantic information.

The PWI metric requires two components to calculate the substring-semantic information: $P(s|Sem(x))$, the substring generation probability, and $\log \frac{P(Sem(x)|s)}{P(Sem(x))}$, the piecewise mutual information between $s$ and $Sem(x)$. Similar to the bag-of-words model, we will calculate the two components using the frequency of $s$ in $\mathbb{X}^p$ and the inverse frequency of $s$ in the corpus $\mathcal{D}$.

$$I(s, Sem(x)) = P(s|Sem(x)) \log \frac{P(Sem(x)|s)}{P(Sem(x))} \qquad (4)$$

## 3.2 CALCULATING PWI USING MAXIMAL SUBSTRINGS

Naively measuring substring frequency among paraphrases $\mathbb{X}^p$ will yield a misleading estimate of $P(s|Sem(x))$. The reason is that one substring can be nested in another substring. If a substring $s$ is generated to convey semantic information, we will observe an occurrence of $s$ along with an

occurrence of all its substrings. Hence, the naive substring frequency will wrongly count substring occurrences caused by the generation of larger substrings as occurrences caused by $P(s|Sem(x))$. Let us consider the example illustrated in Figure 3. All three substrings in the example have a frequency of 2, yet only the first substring carries significant semantic information. This is because the occurrence of the first substring causes the occurrence of the second and third substrings. The true frequency of the second and third substrings should be 0 instead of 2.

We introduce the notion of maximal substring to counter this problem. Given a source sentence $x$ and a paraphrase $x_i^p$, the maximal substring between the two is defined in Equation 5. Intuitively, a maximal substring is the largest substring that occurs in both $x$ and $x_i^p$. Formally, we denote the partial order relationship of string $\alpha$ being a substring in string $\beta$ by $\alpha \leq \beta$, and denote the set of maximal substrings by $MS(x, x_i^p)$. Using maximal substrings, we can avoid over-counting substring occurrences caused by the generation of larger substrings.

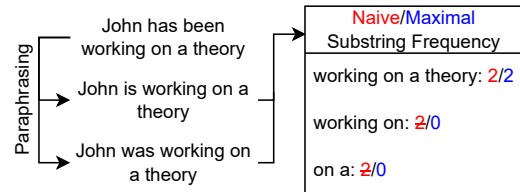

Figure 3: An example for naive substring frequency among paraphrases failing to estimate $P(s|Sem(x))$.

$$MS(x, x_i^p) := \{\alpha : \alpha \leq x \wedge \alpha \leq x_i^p \wedge \forall \alpha'(\alpha < \alpha' \implies \neg \alpha' \leq x \vee \neg \alpha' \leq x_i^p)\} \tag{5}$$

We are now ready to define $P(s|Sem(x))$ using the paraphrasing distribution $P(x^p|Sem(x))$ and the notion of maximal substrings. We define $P(s|Sem(x))$ to be proportional to $s$'s probability of being generated as a maximal substring in paraphrases (Equation 6). The probability can then be approximated using the maximal substring frequency $F(s, \mathbb{X}^p)$, as shown in Equation 7.

$$P(s|Sem(x)) \propto \mathop{\mathbb{E}}_{x_i^p \sim P(x^p|Sem(x))} \mathbf{1}(s \in MS(x_i^p, x)) \tag{6}$$

$$\approx F(s, \mathbb{X}^p) \tag{7}$$

Similarly, we define the inverse document frequency for maximal substrings (Equation 8). The inverse document frequency can serve as an estimate of the piecewise substring-semantics information, quantifying how useful a substring is to convey semantic information. A high inverse document frequency implies that only a few $Sem(x)$ in the corpus generate $s$ as their maximal substring. In other words, we can easily identify the target semantics by examining whether $s$ appears as maximal substrings.

$$\log \frac{P(Sem(x)|s)}{P(Sem(x))} \approx \log \frac{|\mathcal{D}|}{|\{x' : x' \in \mathcal{D} \wedge s \in MS(x, x')\}|} \tag{8}$$

### 3.3 Estimating SemInfo

A constituent structure $t$ can be represented as a set of constituent substrings. We define SemInfo, the information between $t$ and $Sem(x)$, as the cumulative substring-semantics information associated with $t$ (Equation 9). We estimate the substring-semantics information with the maximal substring frequency-inverse document frequency developed in the above section.

$$I(t, Sem(x)) = \sum_{s \in t} I(s, Sem(x)) \tag{9}$$

$$\propto \sum_{s \in t} \underbrace{F(s, \mathbb{X}^p) \log \frac{|\mathcal{D}|}{|\{x' : x' \in \mathcal{D} \wedge s \in MS(x, x')\}|}}_{\text{maximal substring frequency-inverse document frequency}} \tag{10}$$

## 4 SemInfo Maximization via TreeCRF Model

We train our PCFG models on Equation 12 using the pipeline shown in Figure 4. The pipeline consists of three steps: (1) We compute the $\log(Z(x))$ by applying the inside algorithm on the PCFG model. This step yields the leading log-likelihood term in Equation 12. More importantly, it constructs the computation graph needed to calculate the span-posterior probability $P(s \text{ is a constituent}|x)$. (2) We extract the span-posterior probability via back-propagating

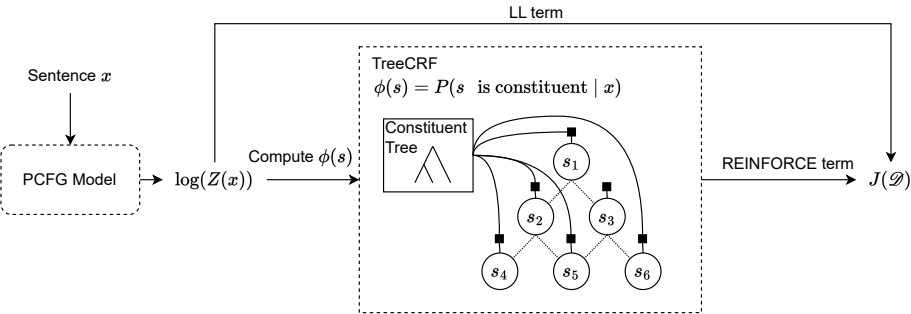

Figure 4: Pipeline of our SemInfo maximization training

$\log(Z(x))$ and parameterize a TreeCRF model by setting $\phi(s, x) = P(s$ is a constituent$|x)$ (Equation 11). This parametrization leads to a tree distribution $P^{CRF}(t|x)$ that functions as an *one-step Reinforcement Learning agent*. (3) We train the PCFG model by applying the SemInfo maximization on $P^{CRF}(t|x)$. This TreeCRF-based training method is equivalent to applying a mean-field SemInfo maximization to the PCFG model. We choose the TreeCRF model because it enables efficient sampling from $P^{CRF}(t|x)$ and entropy calculation for the distribution. As discussed in Appendix A.1, the TreeCRF-based training method performs equivalently to applying SemInfo maximization on the PCFG model directly. Yet, the TreeCRF-based method runs 4x faster and uses $\frac{1}{6}$ the memory compared to the direct PCFG optimization method.

We apply the REINFORCE algorithm with average baseline (Williams, 1992) to facilitate the training. We include the maximum entropy regularization (Ziebart et al., 2008) and the traditional LL term $\log Z(x)$ in the training. Notably, the LL term significantly stabilizes the training process. This stabilization effect may be related to the strong correlation between LL values and parsing accuracy at the early training stage, as discussed in Section 5.3.2.

$$P^{CRF}(t|x) \propto \prod_{s \in t} P(s \text{ is a constituent}|x)$$

$$= \prod_{s \in t} \sum_{A \in NT} \frac{\partial \log Z(x)}{\partial \log \beta(s, A)} \tag{11}$$

$$\mathcal{J}(\mathcal{D}) = \mathbb{E}_{x \sim \mathcal{D}}[\log Z(x) + \mathbb{E}_{t \sim P^{CRF}(t|x)}[\log P^{CRF}(t|x)(I(t, Sem(x)) - \mathbb{E}_{t \sim P^{CRF}(t|x)} I(t, Sem(x)) + \beta H(P(t|x)))]] \tag{12}$$

## 5 EXPERIMENT

### 5.1 EXPERIMENT SETUP

We evaluate the effect of the SemInfo maximization objective on five latest PCFG variants: Neural-PCFG (NPCFG), Compound-PCFG (CPCFG) (Kim et al., 2019a), TNPCFG (Yang et al., 2021b), Simple-NPCFG (SNPCFG), and Simple-CPCFG (SCPCFG) (Liu et al., 2023).[2] SNPCFG and SCPCFG represent the current state-of-the-art for non-ensemble unsupervised constituency parsing. We use 60 NTs for NPCFG and CPCFG, and 1024 NTs for TNPCFG, SNPCFG, and SCPCFG in our experiment. We conduct the evaluations in three datasets and four languages, namely Penn TreeBank (PTB) (Marcus et al., 1999) for English, Chinese Treebank 5.1 (CTB) (Palmer et al., 2005) for Chinese, and SPMRL (Seddah et al., 2013) for German and French. We adopt the standard data split for the PTB dataset (Sections 02-21 for training, Section 22 for validation, and Section 23 for testing) (Kim et al., 2019a). We adopt the official data split for the CTB and SPMRL datasets.

Following Shen et al. (2017), we train the PCFG model on raw text without punctuations and evaluate its parsing performance using the $SF1^c$ scores. When computing the $SF1^c$ score, we aggregate $SF1^i$ only for sentences longer than two words and drop trivial spans (i.e., sentence-level spans

---

[2]Our implementation is based on the source code of Yang et al. (2021b) and Liu et al. (2023)

| | English | | Chinese | | French | | German | |
|---|---|---|---|---|---|---|---|---|
| | SemInfo (Ours) | LL | SemInfo | LL | SemInfo | LL | SemInfo | LL |
| CPCFG | $\mathbf{65.74}_{\pm0.81}$ | $53.75_{\pm0.81}$ | $50.39_{\pm0.87}$ | $51.45_{\pm0.49}$ | $\mathbf{52.15}_{\pm0.75}$ | $47.50_{\pm0.41}$ | $\mathbf{49.80}_{\pm0.31}$ | $45.64_{\pm0.73}$ |
| NPCFG | $\mathbf{64.45}_{\pm1.13}$ | $50.96_{\pm1.82}$ | $\mathbf{53.30}_{\pm0.42}$ | $42.12_{\pm3.07}$ | $\mathbf{52.36}_{\pm0.62}$ | $47.95_{\pm0.09}$ | $\mathbf{50.74}_{\pm0.28}$ | $45.85_{\pm0.63}$ |
| SCPCFG | $\mathbf{67.27}_{\pm1.08}$ | $49.42_{\pm2.42}$ | $51.76_{\pm0.54}$ | $46.20_{\pm3.65}$ | $\mathbf{52.79}_{\pm0.80}$ | $45.03_{\pm0.42}$ | $47.97_{\pm0.76}$ | $45.50_{\pm0.71}$ |
| SNPCFG | $\mathbf{67.15}_{\pm0.62}$ | $58.19_{\pm1.13}$ | $\mathbf{51.55}_{\pm0.82}$ | $43.79_{\pm0.39}$ | $\mathbf{55.21}_{\pm0.47}$ | $49.64_{\pm0.91}$ | $\mathbf{49.65}_{\pm0.29}$ | $40.51_{\pm1.26}$ |
| TNPCFG | $\mathbf{66.55}_{\pm0.96}$ | $53.37_{\pm4.28}$ | $51.79_{\pm0.83}$ | $45.14_{\pm3.05}$ | $\mathbf{54.11}_{\pm0.66}$ | $39.97_{\pm4.10}$ | $\mathbf{49.26}_{\pm0.64}$ | $44.94_{\pm1.34}$ |
| Average $\Delta$ | +13.09 | | +6.02 | | +7.31 | | +4.92 | |
| MaxTreeDecoding | 58.28 | | 49.03 | | 52.03 | | 50.82 | |
| GPT4o-mini | 36.16 | | 11.82 | | 30.01 | | 33.56 | |

Table 1: $SF1^c$ scores of five PCFG variants trained with SemInfo and LL. Each cell in the upper section reports the mean $SF1^c$ score and the standard deviation across three *identical and independently trained* PCFG models. Average $\Delta$ indicates average improvements in the $SF1^c$ score when training with SemInfo compared to LL. Improvements that are statistically significant ($p < 0.05$) are highlighted in bold.

and spans with only one word). We use both the $SF1^c$ and $SF1^i$ scores to evaluate the correlation between the SemInfo value and parsing accuracy.

We use the `gpt-4o-mini-2024-07-18` model as our paraphrasing model and apply the same word normalization techniques as in Chen et al. (2024). The average paraphrasing cost is about 5 USD using OpenAI's batch API. We use eight semantic-preserving prompts for the paraphrasing model.[3] We apply the snowball stemmer (Bird & Loper, 2004) to normalize the source sentence and its paraphrases before calculating the maximal substring frequency and the inverse document frequency. We apply the log-normalization (Sparck Jones, 1972) to the maximal substring frequency to avoid some high-frequency substrings dominating the SemInfo value. The log-normalization is compatible with the PWI framework, which treats the normalization as an optional step to estimate $P(s|Sem(x))$. In preliminary experiments, the log-normalization variant performs marginally but consistently better than the unnormalized variant.

## 5.2 SemInfo Maximization Significantly Improves Parsing Accuracy

Table 1 compares SemInfo-trained PCFGs and LL-trained PCFGs on five contemporary PCFG variants and four languages. For each variant, we independently train three PCFG models on the SemInfo and LL objectives and report the mean and standard deviation of their $SF1^c$ scores. We can observe that most SemInfo-trained PCFGs achieve significantly higher parsing accuracy than their LL-trained counterparts. The average improvements are 13.09, 6.02, 7.31, and 4.92 $SF1^c$ scores in English, Chinese, French, and German, respectively. Two-tailed t-tests indicate the improvement to be statistically significant (p<0.05) in 17 out of 20 combinations. Two of the three insignificant results are due to the high score variance of the LL-trained PCFGs. The significant improvement demonstrates the benefit of the SemInfo maximization objective in the unsupervised constituency parsing task. The result also confirms the importance of semantic factors in identifying the syntactic constituent structure.

Table 1 also compares the SemInfo trained PCFG with two baseline parsers: Maximum Tree Decoding (MTD) parser, which predicts the structure with maximum SemInfo value, and GPT4o-mini parser that asks the GPT4o-mini model to predict the structure in bracket form directly. Among the two baselines, we see that the MTD parser has significantly higher $SF1^c$ scores than the GPT4o-mini parser across the four languages. The accuracy gap indicates that SemInfo is discovering non-trivial information about the constituent structure. Comparing the SemInfo-trained PCFG and the MTD parser, we see that all SemInfo-trained PCFG variants outperform the MTD parser in English, Chinese, and French. The accuracy improvement indicates that the constituent information provided by the SemInfo value is noisy, and the grammar learns to mitigate the noises. We can again confirm PCFG's de-noising effect in an experiment investigating how paraphrasing noise affects parsing performance (Appendix A.5).

In German, SemInfo-trained PCFGs perform worse than the MTD parser. One possible reason is that the German validation/testing set has a significantly different word vocabulary compared to the training set, unlike the datasets in the other three languages. The out-of-vocabulary rate in the German dataset is 14%, while the rate is 5%, 6%, and 7% in the English, Chinese, and French

---

[3]Detailed prompts are listed in Section A.8

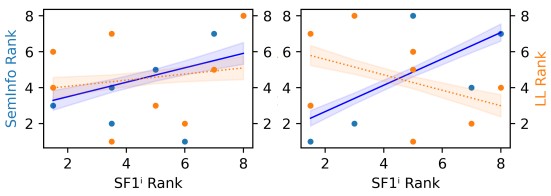

|        | SemInfo-SF1$^i$ | LL-SF1$^i$ | SemInfo-LL |
|--------|-----------------|------------|------------|
| CPCFG  | 0.6518          | 0.0223     | 0.0196     |
| NPCFG  | 0.6347          | -0.0074    | -0.0045    |
| SCPCFG | 0.6431          | -0.0013    | 0.0505     |
| SNPCFG | 0.9289          | 0.0102     | 0.0182     |
| TNPCFG | 0.6449          | 0.1077     | 0.1426     |

Table 2: Spearman correlation coefficient among (SemInfo, LL, SF1$^i$), and LL over the English validation set. Correlations are aggregated at the corpus-level.

Figure 5: Spearman rank analysis of (SemInfo, LL, SF1$^i$) pairs obtained from eight independently trained NPCFG models. The values are measured on two sentences in the English dataset. Please refer to Figure 8 for more examples.

datasets. This shift in word distribution might be a significant factor in German PCFGs' poor parsing accuracy.

## 5.3 SemInfo Strongly Correlates with Parsing Accuracy

In this section, we investigate how the SemInfo and LL functions contribute to obtaining high-quality PCFG parsers from two aspects: (1) Whether the function can accurately evaluate the model's prediction (measured by SF1$^i$). (2) Whether the function can approximately rank PCFG parsers in accordance with parsing performance (measured by SF1$^c$). Our experiments indicate that SemInfo can serve as an accurate estimate of parsing accuracy and that SemInfo is a better training objective for unsupervised parsers than LL. We evaluate the two aspects using the Spearman correlation (Spearman, 1904) between the SemInfo/LL values and the SF1$^i$/SF1$^c$ scores. We refer to the correlation analysis using the SF1$^i$ score *sentence-level* analysis and the analysis using the SF1$^c$ score *corpus-level* analysis.

### 5.3.1 SemInfo Estimates Parsing Accuracy

The sentence-level analysis assesses the SemInfo/LL's capability to evaluate the model prediction accurately. We independently train eight *identical* PCFG models using the LL maximization objective. Each model is trained with a unique random seed for 30k steps. These eight models produce eight (SF1$^i$, SemInfo, LL) tuples for any given sentence, which we use to calculate the sentence-level Spearman correlation coefficient.

Figure 5 illustrates the correlation gap between SemInfo-SF1$^i$ and LL-SF1$^i$ pairs using two sentences in the English validation set. Between the two sentences, the SemInfo-SF1$^i$ pairs exhibit positive correlations while the LL-SF1$^i$ pairs exhibit no apparent correlations. Table 2 confirms the correlation gap using the correlation coefficient aggregated in the corpus level. We perform mean-aggregation using Fisher's Z transformation (Fisher, 1915). The transformation converts the coefficient to a uni-variance distribution and reduces the negative impact of the aggregation caused by the coefficient's skewed distribution (Silver & Dunlap, 1987). In the table, we observe that the aggregated coefficients for the SemInfo-SF1$^i$ pairs range from 0.6-0.9, whereas the aggregated coefficients for the LL-SF1$^i$ correlation center around 0. We can also consistently observe the correlation gap across multiple training stages, as further discussed in Appendix A.3. The consistent correlation gap, on the one hand, suggests that SemInfo can serve as an accurate estimate of parsing accuracy and that SemInfo is a better training objective for unsupervised parsers than LL. On the other hand, it highlights SemInfo's ability to capture constituent information, reaffirming a close relationship between constituent structure and sentence semantics.

### 5.3.2 SemInfo Ranks PCFG Models Better than LL

The corpus-level analysis evaluates the SemInfo/LL's capability to rank PCFG parsers by their parsing performance. We examine the correlation using model checkpoints collected over different training stages of the above eight PCFG models. Each stage is represented by a window over the amount of training steps. For example, a stage [1k, 10k] contains checkpoints from 1k to 10k steps.

|  | English | Chinese | French | German |
|---|---|---|---|---|
| NPCFG (60NT) | $63.62_{\pm1.07}$ | $\mathbf{53.92}_{\pm0.48}$ | $51.88_{\pm0.73}$ | $47.77_{\pm0.26}$ |
| SCPCFG (1024NT) | $\mathbf{66.92}_{\pm0.76}$ | $52.26_{\pm0.41}$ | $52.29_{\pm0.53}$ | $45.32_{\pm0.67}$ |
| SNPCFG (1024NT) | $66.84_{\pm0.53}$ | $52.04_{\pm0.93}$ | $\mathbf{54.37}_{\pm0.10}$ | $47.27_{\pm0.16}$ |
| Spanoverlap (Chen et al., 2024) | 52.9 | 48.7 | 48.5 | $\mathbf{49.5}$ |
| SCPCFG (2048NT) (Liu et al., 2023) | 60.6 | 42.9 | 49.9 | 49.1 |
| SNPCFG (4096NT) (Liu et al., 2023) | 65.1 | 39.9 | 38 | 46.7 |
| URNNG (Kim et al., 2019b) | 40.7 | 29.1 | - | - |
| NBL-PCFG (Yang et al., 2021a) | 60.4 | - | - | - |
| S-DIORA (Xu et al., 2021) | 57.6 | - | - | - |
| Constituency Test (Cao et al., 2020) | 62.8 | - | - | - |

Table 3: SF1$^c$on English, Chinese, French, and German test sets. The top section shows the score for SemInfo-trained PCFGs while the bottom section shows the result from previous work.

These checkpoints produce a set of (SF1$^c$, corpus-averaged SemInfo, corpus-averaged LL) tuples, which we use to calculate the corpus-level coefficient at that training stage.

Figure 6 illustrates the SemInfo-SF1$^c$ and LL-SF1$^c$ correlation curves for NPCFG.[4] We can observe that LL does have a strong corpus-level correlation with SF1$^c$ at the early stage of training despite having a near-non-existent sentence-level correlation. However, LL's coefficient quickly diminishes as training progresses, dropping below 0.4 at the late training stage. This result indicates that LL identifies a reasonable PCFG parser among a set of poorly performing parsers in the early training stage, explaining why the LL-training can result in non-trivial PCFG parsers despite having negligible correlation in the sentence-level analysis. Yet, this ability quickly degrades as the training progresses. In comparison, SemInfo maintains a strong correlation across the whole training process, which indicates SemInfo's superior capability in ranking PCFG parsers by their performance.

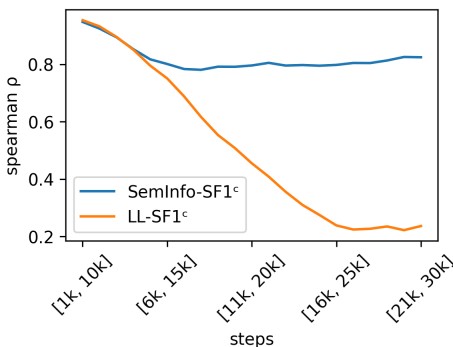

Figure 6: Spearman $\rho$ with SF1$^c$ in different training stages of NPCFG.

### 5.4 COMPARING WITH STATE-OF-THE-ARTS

Table 3 compares three SemInfo-trained PCFG variants with the state-of-the-art non-ensemble methods for unsupervised constituency parsing. The SemInfo-trained PCFGs achieved state-of-the-art level parsing accuracy in English, Chinese, and French, outperforming the second-best algorithm by 1.82, 11.02, and 4.47 SF1$^c$scores, respectively. The SemInfo-trained PCFGs, while using less than half the parameters, perform on par or significantly better than the larger SCPCFG and SNPCFG reported by Liu et al. (2023). The comparison showcases the strong parsing accuracy of the SemInfo-trained PCFGs, confirming the usefulness of semantic information in discovering the constituent structure.

### 6 RELATED WORKS

**Parsing with PCFG** Unsupervised PCFG training is a long-established (Klein & Manning, 2002) and state-of-the-art (Liu et al., 2023) approach for non-ensemble unsupervised constituency parsing. Much research has been dedicated to improving PCFG training from the model perspective, such as scaling up the PCFG model (Yang et al., 2021b; Liu et al., 2023), integrating lexical information (Yang et al., 2021a), and allowing PCFG rule probabilities to condition on sentence embeddings through variational inference (Kim et al., 2019a). Our improvement is from the model optimization perspective and can be combined with the above efforts. Our experiments validate the effectiveness

---

[4]We include the correlation curve for the other four PCFG variants in Appendix A.4.

of the SemInfo maximization objective in improving unlexicalized PCFGs. The SemInfo maximization objective is also applicable to lexicalized PCFGs, which we leave to future work.

**Parsing with Semantics** Zhao & Titov (2020) and Zhang et al. (2021) have sought to improve PCFG training by learning to identify visual features, maximizing the association between constituent structures and these visual features. If we consider the visual features as semantic representations, their approach is effectively maximizing the semantic information of the constituent structure. In comparison, our method shares the same underlying principle but represents the semantics with textual features. Our method leverages large language models as semantic processors, utilizing their outstanding semantic processing capabilities (Minaee et al., 2024). We believe that combining both textual and visual semantic representations presents a significant research direction for unsupervised parsing tasks.

**Improving Parsing with Ensemble Models** Ensembling unsupervised parsers (Shayegh et al., 2024) significantly improves accuracy for unsupervised parsing by aggregating predictions from various base parsers. They show that those base parsers predict the constituent structure differently and utilize the difference to obtain a more accurate parsing result. Our method can be combined with the ensemble method for better parsing accuracy. We conduct a parser agreement analysis in Appendix A.6 to show the potential. The agreement analysis shows an agreement score of 80 among our SemInfo-trained PCFG parsers using various paraphrasing models. The agreement score is similar to that of homogeneous parsers reported in Shayegh et al. (2024). The analysis also shows that our parsers have an agreement score of 50 with other base parsers, similar to the reported score between heterogeneous parsers. The similarity in agreement score suggests that our parsers should be able to serve as a useful component in the ensemble method.

## 7 Conclusion

In this paper, we proposed and validated SemInfo maximization as a novel objective for unsupervised constituency parsing. We developed a bag-of-substrings model to represent the sentence semantics and applied the probability-weighted information metric to estimate the SemInfo. We applied the SemInfo maximization objective to training PCFG parsers. Experiments showed that SemInfo has a strong sentence-level correlation with parsing accuracy and that SemInfo maintains a consistent corpus-level correlation throughout the PCFG training process. These correlation analyses indicate that SemInfo is an accurate estimate of parsing accuracy and that it is a reliable training objective for unsupervised parsers. As a result, SemInfo-trained PCFGs significantly outperformed LL-trained PCFGs across four languages, achieving state-of-the-art level performance in three of them. Our findings highlight the effectiveness of leveraging semantic information in unsupervised constituency parsing, paving the way for semantically-informed unsupervised parsing methods.

## 8 Reproducibility

We provide detailed explanation of our method in Sections 3 and 4. We outline further implementation details, such as the data source, model architecture, and hyper-parameter settings, in Section 5.1. We release the source code at https://github.com/junjiechen-chris/Improving-Unsupervised-Constituency-Parsing-via-Maximizing-Semantic-Information.git.

## 9 Acknowledgment

This research was funded by the Japan Society for the Promotion of Science through the Research Fellowships for Young Scientists (Grant No. JP23KJ0565) and by the KAKEN project (Grant No. 24H00087). We sincerely thank them for their financial support of the research. We also appreciate the reviewer's thorough evaluation and valuable suggestions during the review process.

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

| Strategy | RL Baseline | Sampling distribution | Learning Strategy |
|---|---|---|---|
| Action-V | V-function | Action distribution | SemInfo maximization |
| Posterior-V | V-function | Tree posterior distribution | SemInfo maximization |
| Posterior-Avg | Sample average | Tree posterior distribution | SemInfo maximization |
| Supervised | - | - | Supervised learning |
| LL | - | - | Likelihood maximization |

Table 4: Lookup table for optimization strategies and detailed descriptions

---

**Algorithm 1** TreeCRF Sampler

---

1: **function** CRF-Sampler($i, j, x$)
2:   **if** $j = i + 1$ **then**
3:     **Return** leaf node $(i, j)$
4:   **else**
5:     Sample split index $k \sim \pi^{CRF}(k \mid (i, j))$ following Equation 14 Johnson et al. (2007)
6:     $T_{\text{left}} \leftarrow$ CRF-Sampler($i, k, x$)
7:     $T_{\text{right}} \leftarrow$ CRF-Sampler($k, j, x$)
8:     **Return** node $(i, j)$ with children $T_{\text{left}}$ and $T_{\text{right}}$
9:   **end if**
10: **end function**

---

# A  APPENDIX

## A.1  ADVANCED SEMINFO MAXIMIZATION

In Section 4, we presented a SemInfo maximization method that performs mean-field optimization through a TreeCRF model. This method parameterizes a TreeCRF model using the span-posterior probability and maximizes the expected SemInfo value of the TreeCRF distribution. While our study demonstrated significant accuracy improvement by the TreeCRF-based SemInfo maximization training, it raises a new question: What is the advantage of optimizing the PCFG parameters through the TreeCRF model compared to optimizing those parameters directly? In the experiment presented in this section, we found no significant difference between the TreeCRF-based and PCFG-based optimizations. It shows that the demonstrated accuracy improvement does not depend on particular optimization methods, highlighting the contribution of the SemInfo maximization objective to accurate unsupervised parsing. In addition, we found that the TreeCRF-based optimization is more time and space-efficient than the PCFG-based optimization, which makes the TreeCRF-based optimization preferable.

As shown in Table 4, we compare three SemInfo maximization strategies combined with two optimization methods (TreeCRF-based and PCFG-based methods). We include an LL-trained parser and a supervised parser as baselines. The two baselines serve as the lower and upper bounds for the comparison, respectively.

### A.1.1  TREE POSTERIOR AND SAMPLING DISTRIBUTIONS

Both the TreeCRF-based and PCFG-based optimizations aim to maximize the expected SemInfo with on-policy Reinforcement Learning (RL), but they operate on two policy distributions: the TreeCRF-based posterior distribution $P^{CRF}(t|x)$ (Equation 11) and the PCFG-based posterior distribution $P^{PCFG}(t|x)$ (Equation 13). Sampling from both distributions involves multiple span-splitting steps (Algorithm 1 for the TreeCRF model and Algorithm 2 for the PCFG model). The sampler starts with the sentence-level span $((1, n)$ for the TreeCRF model; $(S, 1, n)$ for the PCFG model) and recursively makes splitting decisions $((i, j) \rightarrow (i, k)(k, j)$ for the TreeCRF model; $(A, i, j) \rightarrow (B, i, k)(C, k, j)$ for the PCFG model). The sampler repeats this span-splitting process until it reaches single-word spans $(j = i + 1)$.

$$P^{PCFG}(t|x) = \frac{P(x, t)}{\sum_t P(x, t)} \tag{13}$$

---

**Algorithm 2** PCFG Sampler

---

1: **function** PCFG-Sampler$(A, i, j, x)$
2: **if** $j = i + 1$ **then**
3:     **Return** leaf node $(A, i, j)$
4: **else**
5:     Sample split index $B, C, k \sim \pi^{PCFG}(B, C, k \mid (A, i, j))$ following Equation 15
6:     $T_{\text{left}} \leftarrow$ PCFG-Sampler$(B, i, k, x)$
7:     $T_{\text{right}} \leftarrow$ PCFG-Sampler$(C, k, j, x)$
8:     **Return** node $(A, i, j)$ with children $T_{\text{left}}$ and $T_{\text{right}}$
9: **end if**
10: **end function**

---

$$\pi^{CRF}(k|(i,j)) = \frac{\exp(\sum_{s \leq (i,k)} \log P(s \text{ is a const.}|x) + \sum_{s \leq (k,j)} \log P(s \text{ is a const.}|x))}{\sum_k \exp(\sum_{s \leq (i,k)} \log P(s \text{ is a const.}|x) + \sum_{s \leq (k,j)} \log P(s \text{ is a const.}|x))} \tag{14}$$

$$\pi^{PCFG}(B, C, k|(A, i, j)) = \frac{P(A \to BC)\beta(B, i, k)\beta(C, k, j)}{\beta(A, i, j)} \tag{15}$$

### A.1.2    THREE SEMINFO MAXIMIZATION STRATEGIES

In this subsection, we introduce two RL optimization methods: posterior and action optimizations. We further combine the optimization methods with two RL baseline estimations: average baseline and $V$-function baselines. Since the average baseline cannot be applied to the action optimization, the combination of the optimizations and baselines gives three optimization strategies (Table 4).

The posterior optimization is similar to the method explained in the main text: (1) sampling tree from either $P^{CRF}(t|x)$ or $P^{PCFG}(t|x)$; and (2) perform policy gradient optimization in accordance with Equation 12. In contrast, the action optimization considers the tree sampling process as an RL trajectory and applies the SemInfo maximization through $\pi^{CRF}$ or $\pi^{PCFG}$. The two optimizations differ in how the RL agent is defined. The posterior optimization defines the RL agent as a one-step agent and seeks to maximize the SemInfo values for the parser-predicted trees. The action optimization defines the RL agent as a span-splitting agent and seeks to maximize the SemInfo for the tree resulting from the span-splitting decision.

### A.1.3    $V$-FUNCTION COMPUTATION

In the PCFG setting, the $V$-function can be computed precisely and efficiently using dynamic programming. The $V$-function estimates the expected return of visiting a state $s$ ($s = (i, j)$ for the TreeCRF model and $s = (A, i, j)$ for the PCFG model) using a policy $\pi$ (Equation 16). It enables the estimation of the advantage function $A(s, a)$, which evaluates how effective $a$ is in maximizing the SemInfo value of the sampled tree (Equation 17). Algorithm 3 and Algorithm 4 details the $V$-function computation for both the TreeCRF and PCFG-based optimizations. The two algorithms share the same backbone but differ in the span-splitting agent $\pi$.

$$V(s) = \mathbb{E}_{(s_0, a_0, s_1, a_1, \dots) \sim \pi} \left[ \sum_{(s_i, a_i)} r(s_i, a_i) \right] \tag{16}$$

$$A(s, a) = r(s, a) + V(s') - V(s) \tag{17}$$

### A.1.4    TREECRF-BASED OPTIMIZATIONS

Equation 18 (restatement of Equation 12), 19, and 20 details the training objective for the TreeCRF-based Posterior-Avg, Posterior-V, and Action-V strategies, respectively. Comparing Equation 18 and Equation 19, the Posterior-V improves over the Posterior-Avg by substituting the sample-average baseline with the $V$-function baseline. Our preliminary experiments indicate that the substitution results in faster convergence and slightly higher parsing accuracy. The Action-V strategy directly

---

**Algorithm 3** TreeCRF-based V-Function Computation

---

**Require:** Subtree-selection agent $\pi^{CRF}(i,j)$ governing all possible split decisions of span $(i,j) \rightarrow (i,k)(k,j)$.
**Require:** SemInfo function $r(i,j) = I(x_{i:j}, Sem(x))$.
**Ensure:** $V(i,j)$ for all spans.
1: Initialize $V(i,j) = 0$ for all spans $(i,j)$.
2: $w \leftarrow 2$
3: **repeat**
4:     **for** each span $s$ of length $w$ **do**
5:         $V(s) \leftarrow \begin{cases} 0 & w = 1 \\ r(i,j) & w = 2 \\ \mathbb{E}_{\pi^{CRF}(k|s)}\left[(V(i,k) + V(k,j))\right] + r(i,j) & w > 2 \end{cases}$
6:     **end for**
7:     $w \leftarrow w + 1$
8: **until** $w = n$
9: **return** $V$

---

**Algorithm 4** PCFG-based V-Function Computation

---

**Require:** Subtree-selection agent $\pi^{PCFG}(A,i,j)$ governing all possible split decisions of span $(A,i,j) \rightarrow (B,i,k)(C,k,j)$.
**Require:** SemInfo function $r(A,i,j) = I(x_{i:j}, Sem(x))$.
**Ensure:** $V(A,i,j)$.
1: Initialize $V(A,i,j) = 0$ for all $(A,i,j)$.
2: $w \leftarrow 2$
3: **repeat**
4:     **for** each span $s$ of length $w$ **do**
5:         $V(s) \leftarrow \begin{cases} 0 & w = 1 \\ r(A,i,j) & w = 2 \\ \mathbb{E}_{\pi^{PCFG}(B,C,k|A,i,j)}\left[(V(B,i,k) + V(C,k,j))\right] + r(A,i,j) & w > 2 \end{cases}$
6:     **end for**
7:     $w \leftarrow w + 1$
8: **until** $w = n$
9: **return** $V$

---

optimizes $\pi^{CRF}$ using the advantage function $A$, which we further augment with the Generalized Advantage Estimation Schulman et al. (2016).

$$\mathcal{J}(\mathcal{D}) = \mathbb{E}_{x \sim \mathcal{D}}[\log Z(x) + \mathbb{E}_{t \sim P^{CRF}(t|x)}[\log P^{CRF}(t|x)(I(t, Sem(x)) - \mathbb{E}_{t \sim P^{CRF}(t|x)} I(t, Sem(x)) + \beta H(P(t|x)))]] \tag{18}$$

$$\mathcal{J}(\mathcal{D}) = \mathbb{E}_{x \sim \mathcal{D}}[\log Z(x) + \mathbb{E}_{t \sim P^{CRF}(t|x)}[\log P^{CRF}(t|x)(I(t, Sem(x)) - V_x(1,n) + \beta H(P^{CRF}(t|x)))]] \tag{19}$$

$$\mathcal{J}(\mathcal{D}) = \mathbb{E}_{x \sim \mathcal{D}}[\log Z(x) + \mathbb{E}_{((1,n),k,\ldots) \sim \pi^{CRF}}[\sum_{(i,j),k} \log \pi^{CRF}(k|i,j)(A(i,j,k) + \beta H(\pi^{CRF}(k|i,j)))]] \tag{20}$$

### A.1.5 PCFG-BASED OPTIMIZATIONS

Equation 21, 22, and 23 details the training objective for the PCFG-based Posterior-Avg, Posterior-V, and Action-V strategies, respectively. These strategies are defined similarly to the TreeCRF-based strategies but replace the TreeCRF-based distributions with the PCFG-based distributions.

|  | NPCFG | | SNPCFG | |
|---|---|---|---|---|
|  | TreeCRF | PCFG | TreeCRF | PCFG |
| Action-V | 65.16±1.15 | 62.66±1.76 | 67.21±0.33 | 66.04±0.38 |
| Posterior-V | 66.82±0.32 | 66.37±1.71 | 67.87±0.54 | 66.32±1.39 |
| Posterior-Avg | 65.55±0.75 | 65.64±1.34 | 66.77±0.14 | 66.85±0.32 |
| Supervised | 69.05±0.55 | 73.54±0.11 | 71.83±0.21 | 74.78±0.23 |
| LL | 53.34±0.59 | | 57.84±2.61 | |

Table 5: $SF1^c$ scores of two PCFG variants trained combined with three SemInfo maximization strategies. We retrained the TreeCRF-based Posterior-Avg model and the LL model in this experiment.

$$\mathcal{J}(\mathcal{D}) = \mathop{\mathbb{E}}_{x \sim \mathcal{D}}[\log Z(x) + \mathop{\mathbb{E}}_{t \sim P^{PCFG}(t|x)}[\log P^{PCFG}(t|x)(I(t, Sem(x)) - \mathop{\mathbb{E}}_{t \sim P^{PCFG}(t|x)} I(t, Sem(x)))]] \tag{21}$$

$$\mathcal{J}(\mathcal{D}) = \mathop{\mathbb{E}}_{x \sim \mathcal{D}}[\log Z(x) + \mathop{\mathbb{E}}_{t \sim P^{PCFG}(t|x)}[\log P^{PCFG}(t|x)(I(t, Sem(x)) - V_x(S, 1, n))]] \tag{22}$$

$$\mathcal{J}(\mathcal{D}) = \mathop{\mathbb{E}}_{x \sim \mathcal{D}}[\log Z(x) + \mathop{\mathbb{E}}_{((A,1,n),(B,C,k),\dots) \sim \pi^{PCFG}}[ \sum_{(A,i,j),(B,C,k)} \log \pi^{PCFG}(B, C, k|A, i, j)A(A, i, j, B, C, k)]] \tag{23}$$

### A.1.6 RESULT

Table 5 evaluates the TreeCRF and PCFG-based optimization methods using two PCFG variants. Both methods yield parsers of similar performance. The TreeCRF-based method yields higher mean parsing accuracy than the PCFG-based method, yet the difference is within the margin of error. All combinations yield parsers with 65 67$SF1^c$ scores, except for the NPCFG+Action-V+PCFG-based optimization combination. This combination results in parsers with 62.66 mean $SF1^c$ score, massively underperforming other combinations. The underperformance might be related to the low model capacity of the PCFG model, as we did not observe similar performance degradation in the SNPCFG model and other high-capacity models tested in our preliminary experiment. Overall, the comparison disentangled the high accuracy of the SemInfo-trained PCFGs from specific optimization algorithms. It highlights the contribution of the SemInfo maximization objective to accurate unsupervised parsing.

In comparison with the supervised baseline, all combinations yield parsers with accuracy within 8 $SF1^c$ scores from the supervised baseline. The small gap, on the one hand, showcases the strong performance of the SemInfo-trained PCFG parsers. On the other hand, it indicates that the PCFG model might limit further development of semantic-aware unsupervised parsers. A more expressive parsing model (e.g., the Recurrent Neural Network Model Dyer et al. (2016)) might be necessary in future studies.

While the TreeCRF-based and PCFG-based methods yield parsers of similar accuracies, we found the TreeCRF-based optimization more time and space-efficient than the PCFG-based optimization. The TreeCRF-based optimization trains NPCFG parsers at 4x the speed and uses only $\frac{1}{6}$ the memory. It also trains SNPCFG parsers at 8x the speed and uses $\frac{1}{8}$ the memory. The improved training efficiency enables the further scaling of the PCFG model.

### A.2 COMPUTING SPAN-POSTERIOR PROBABILITY VIA BACK-PROPAGATION

This section explains how the span-posterior probability $P(s \text{ is a constituent}|x)$ is computed using back-propagation.

$$P(s \text{ is a constituent}|x) = \sum_{A \in NT} \frac{\partial \log Z(x)}{\partial \log \beta(s, A)} \tag{24}$$

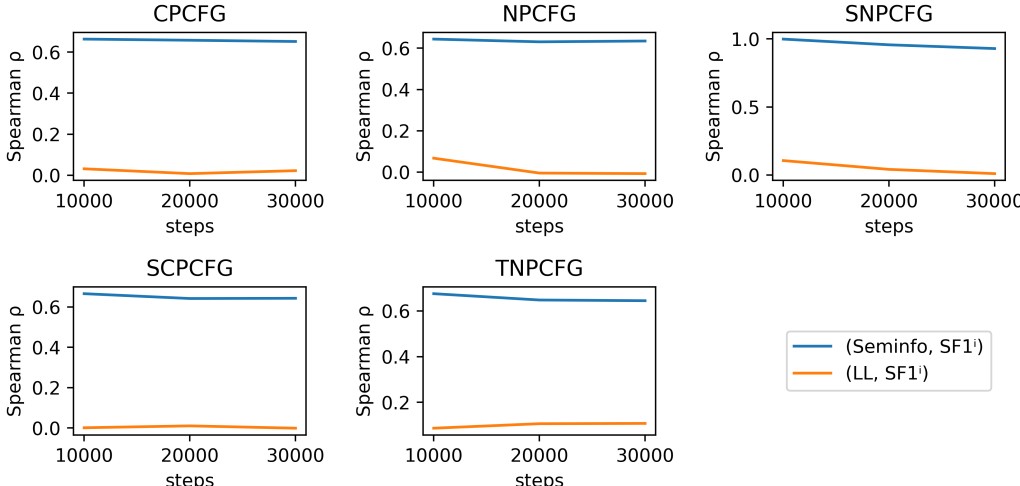

Figure 7: Sentence-level Spearman correlations for models trained for 10k steps, 20k steps, and 30k steps.

*Proof.* Firstly, we define the span-posterior probability as Equation 25. Here $s$ is a substring of $x$, spanning from the $i$-th word to the $j$-th word (i.e., $s := (x_i, ..., x_j)$). Intuitively, $s$ is a constituent if there exists a non-terminal $A$ that expands into $s$.

$$P(s \text{ is a constituent}|x) = \frac{\sum_{A \in NT} P(S \to x \land A \to s_{i,j})}{P(x)} \qquad (25)$$

We split $P(S \to x \land A \to s_{i,j})$ into two parts in Equation 26: $P(S \to x_1, ..., x_{i-1}, A, x_{j+1}, ...)$, the probability of generating words *outside* $s$, and $P(A \to s)$, the probability generating words *inside* $s$. The outside probability can be computed using back-propagation (Eisner, 2016). The inside probability is already computed by the $\beta$ table. Exploiting algebraic transformations shown in Equation 28, we can derive the formula shown in Equation 24.

$$P(s \text{ is a constituent}|x) = \frac{1}{Z(x)} \sum_{A \in NT} P(S \to x_1, ..., x_{i-1}, A, x_{j+1}, ...) P(A \to s) \qquad (26)$$

$$= \frac{1}{Z(x)} \sum_{A \in NT} \frac{\partial Z(x)}{\partial \beta(s, A)} \beta(s, A) \qquad (27)$$

$$= \frac{1}{Z(x)} \sum_{A \in NT} Z(x) \frac{\partial \log Z(x)}{\partial \log \beta(s, A)} \frac{1}{\beta(s, A)} \beta(s, A) \qquad (28)$$

$$= \sum_{A \in NT} \frac{\partial \log Z(X)}{\partial \log \beta(s, A)} \qquad (29)$$

$\square$

### A.3 SENTENCE-LEVEL CORRELATION IN DIFFERENT TRAINING STAGES

In Table 2, we showed a strong sentence-level correlation between SemInfo and $SF1^i$ but a weak correlation between LL and $SF1^i$. Nevertheless, it remains unclear whether the correlation gap is related to the number of training steps Figure 7 excludes the number of training steps as a factor in the correlation gap. In this experiment, we calculate the correlation coefficient for models trained for 10k steps, 20k steps, and 30k steps. We can observe that, for all PCFG variants, the correlation coefficients for (SemInfo, $SF1^i$) are consistently over 0.6, while the coefficients for (LL, $SF1^i$) are consistently below 0.1. This result underscores our conclusion that SemInfo can serve as an accurate estimate of parsing accuracy.

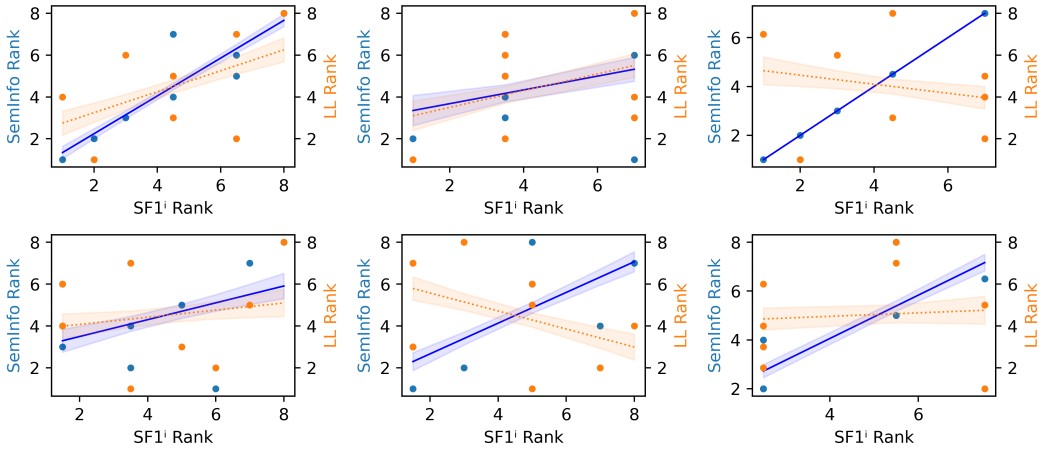

Figure 8: Sentence-level correlation on six random sentences.

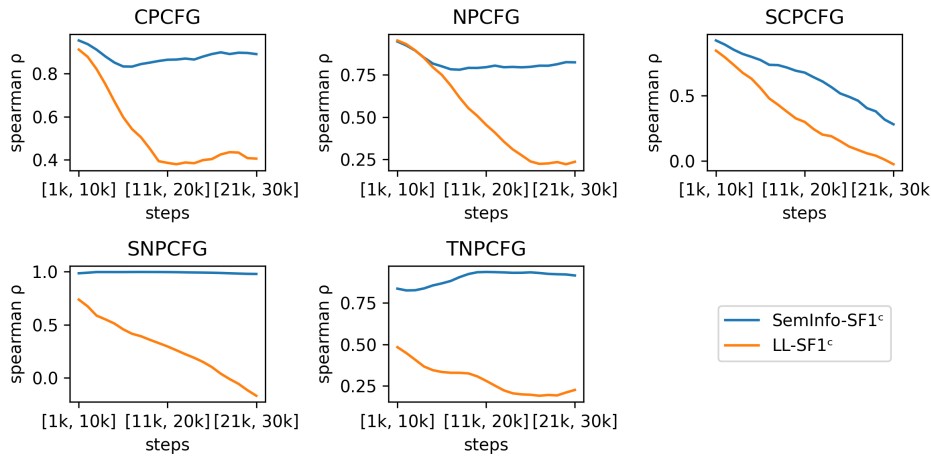

Figure 9: Corpus-level Spearman correlation in different training stages.

## A.4 MORE DETAILED ANALYSIS FOR CORPUS-LEVEL CORRELATION

Figure 9 shows the corpus-level correlation in different training stages for all five PCFG variants. We observe the same phenomenon explained in Section 5.3.2 for CPCFG, NPCFG, SNPCFG, and TNPCFG. The correlation coefficients for (SemInfo, $SF1^c$) are consistently above 0.75, whereas the coefficients for (LL, $SF1^c$) drop quickly as the training progresses. We can observe the stronger correlation between SemInfo and $SF1^c$ in Figure 10. The figure plots the training curves of the corpus-level $SF1^c$ score, the average SemInfo value, and the average LL value over the English validation set. For example, we can see that SemInfo ranks the NPCFG models represented by the green and grey lines as the lowest and those represented by the purple and blue lines as the highest. This largely agrees with the $SF1^c$ scores, where the NPCFG models represented by the green and grey lines are among the bottom three worst-performing models, and the models represented by the blue and purple lines are among the top three best-performing models. In comparison, we see that all models have similar LL scores, which indicates LL's inability to rank models in accordance with their parsing performance. These results underscore our conclusion that SemInfo ranks PCFG models better than LL.

In Figure 9, we observe that the correlation strength for (SemInfo, $SF1^c$) also drops as training progresses in SCPCFG. One reason is that SCPCFG fails to explore constituent structures with high SemInfo values. As shown in Figure 10, the average SemInfo value across the eight models is around 42 for SCPCFG, while the average SemInfo value is greater or equal to 45 for the other

| | Paraphrasing Model Variations | | | | | | |
| --- | --- | --- | --- | --- | --- | --- | --- |
| | Large Models | | | Medium Models | | Small Models | |
| | gpt35 | gpt4o | gpt4omini | llama3.2-3b | qwen2.5-3b | llama3.2 1b | qwen2.5-0.5b |
| SemInfo-NPCFG | 66.85±0.25 | 65.19±0.54 | 64.45±1.13 | 63.78±0.55 | 63.58±0.13 | 63.10±0.70 | 59.01±0.24 |
| SemInfo-MTD | 55.56 | 59.45 | 58.28 | 55.17 | 55.03 | 48.5 | 43.3 |
| LL-NPCFG | 50.96±1.82 | | | | | | |
| Right Branching | 38.4 | | | | | | |

Table 6: SF1$^c$ of the NPCFG and MaxTreeDecoding (MTD) parsers using SemInfo values obtained from seven paraphrasing models. LL-NPCFG indicates the SF1$^c$ score of the LL-trained NPCFG parser.

four PCFG variants. This result indicates that the constituent information provided in low SemInfo regions might contain more noise than the information provided in high SemInfo regions.

## A.5 ROBUSTNESS AGAINST PARAPHRASING NOISES

Table 6 compares the parsing accuracy of NPCFG models trained using seven paraphrasing models. These models are split into three groups: large models (`gpt4o`, `gpt-4o-mini`, `gpt-3.5`), medium models (`llama3.2-3b` and `qwen2.5-3b`), and small models (`llama3.2-1b` and `qwen2.5-0.5b`), each representing paraphrasing models with different levels of noises. The table also includes a MaximumTreeDecoding (MTD) parser, an LL-trained NPCFG parser, and a trivial right-branching parser for reference. We use the MTD parser to reflect the paraphrasing quality because its parsing accuracy depends solely on the paraphrasing quality.

We can observe that the SemInfo-trained NPCFG parsers are robust against paraphrasing noises. The accuracy gap between the best (`gpt4o`) and the worst (`qwen2.5-0.5b`) performing MTD parser is 16.15 SF1$^c$ score. In comparison, the gap between the best and worst performing SemInfo-trained NPCFG parser is 7.84 SF1$^c$ score, less than half of the gap in the MTD parser. In addition, we can observe that the PCFG parser can benefit from the SemInfo maximization training, even when using noisy paraphrases. All SemInfo-trained PCFG parsers significantly outperform their LL-trained counterparts by a large margin. When trained with the most noisy paraphrasing model (`qwen2.5-0.5b`), the SemInfo-trained PCFG parser outperforms its LL-trained counterpart by 9 points. The result suggests that the PCFG model effectively suppresses the paraphrasing noise, leading to robust PCFG parsers.

## A.6 POTENTIAL FOR ENSEMBLING

Figure 12, and Figure 13 suggests that the SemInfo-trained PCFG would benefit from parser ensembling (Shayegh et al., 2024). Shayegh et al. (2024) shows that homogeneous unsupervised parsers (same parser model, different initializations) make mildly distinctive predictions, and heterogeneous parsers (different parser models) make considerably distinctive predictions. Ensembling the parsing results from those parsers effectively suppresses parsing errors made by individual parsers, leading to significant accuracy improvement.

In this section, we evaluate whether our SemInfo-trained PCFGs can benefit from parser ensembling by examining the parser agreement scores for our parser and comparing the score with those reported in Shayegh et al. (2024). If our parser exhibits similar agreement scores, we can consider that our parser would benefit from the parser ensembling. We evaluate the agreement score of our parser and six previous heterogeneous parsers (`CPCFG`, `Constest`, `ContextDistort`, `DIORA`, `NPCFG`, and `SDIORA`).

Figure 12 illustrates the agreement score among parsers using different paraphrasing models. The agreement scores (70-83) are similar to the reported score between homogeneous parsers (74-75 (Shayegh et al., 2024)). This similarity in score suggests that the SemInfo-trained PCFG parsers would benefit from ensembling parsers using various paraphrasing models.

Figure 13 illustrates the agreement score among SemInfo-trained PCFG parsers and previous heterogeneous parsers. We can observe that the agreement score between our SemInfo-trained PCFG parsers and previous parsers ranges from 54-58 (shown in the top-right corner of Figure 13). The score falls in the same range as the score among those previous parsers (46-61, shown in the top-left

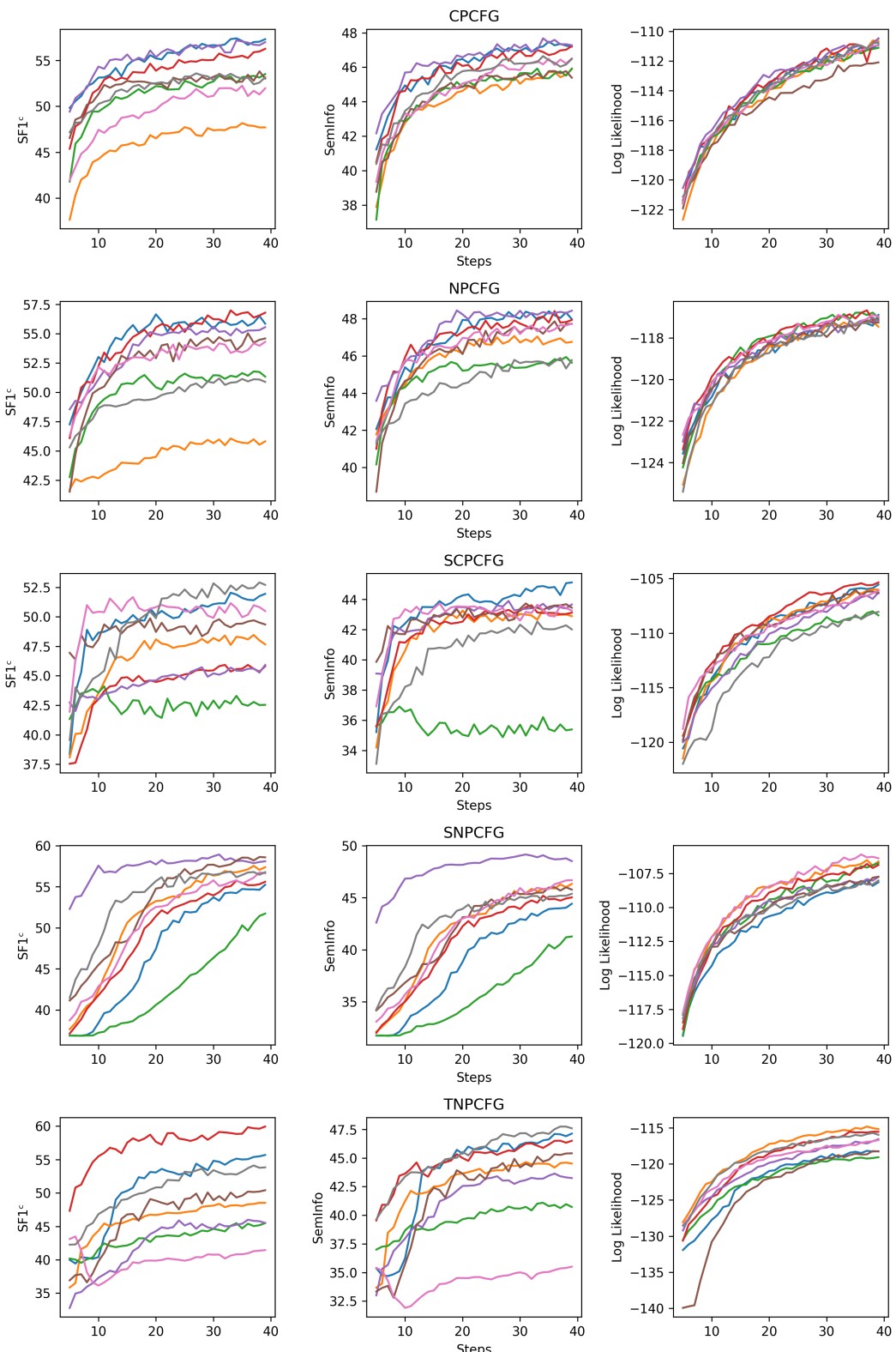

Figure 10: Training curves of SemInfo, LL, and SF1$^c$. Each line represents the curve for a single PCFG model.

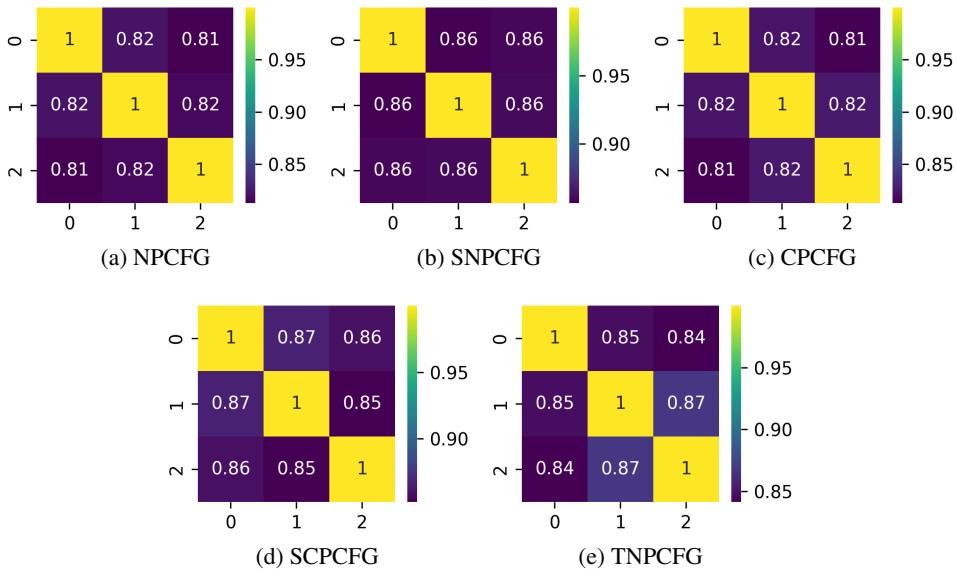

Figure 11: PCFG agreements between independent training runs.

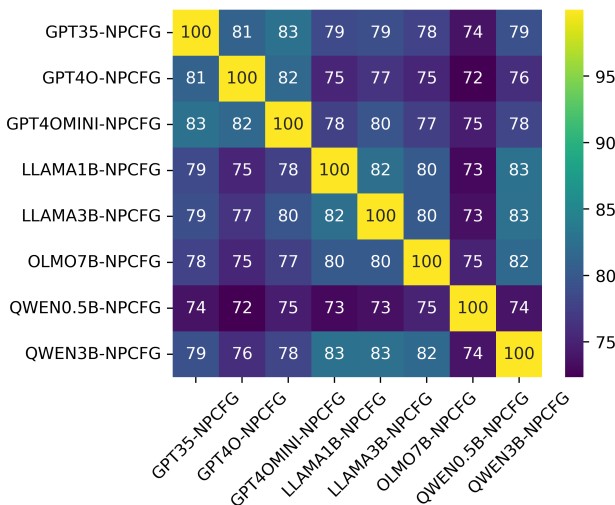

Figure 12: NPCFG parser agreement when trained with different paraphrasing models

corner of Figure 13). Both scores are similar to the reported heterogeneous agreement score (55-63 (Shayegh et al., 2024)). This similarity in score suggests that the SemInfo-trained PCFG parsers would benefit from being ensembled with previous heterogeneous parsers.

## A.7  RECALL ON SIX MOST-FREQUENT CONSTITUENT TYPES

Table 7 shows the recall of the six most frequent constituent types on the English test set, following Yang et al. (2021b). We see that PCFGs trained with SemInfo achieve significant improvement in Noun Phrases (NP), Verb Phrases (VP), and Subordinate Clauses (SBAR). These three constituents are the most typical constituents that carry semantic information. The significant improvement underscores the importance of semantic information in identifying the constituent structure.

| | CPCFG | | NPCFG | | SCPCFG | | SNPCFG | | TNPCFG | | Δ by Type |
|---|---|---|---|---|---|---|---|---|---|---|---|
| | SemInfo (Ours) | LL | SemInfo | LL | SemInfo | LL | SemInfo | LL | SemInfo | LL | |
| NP | $88.88_{\pm0.06}$ | $79.77_{\pm1.58}$ | $88.98_{\pm0.34}$ | $80.63_{\pm2.10}$ | $87.45_{\pm1.16}$ | $79.41_{\pm1.47}$ | $86.51_{\pm0.18}$ | $70.95_{\pm1.64}$ | $87.89_{\pm1.23}$ | $77.73_{\pm5.72}$ | +10.90 |
| VP | $71.19_{\pm1.10}$ | $40.79_{\pm1.49}$ | $65.69_{\pm2.06}$ | $28.29_{\pm3.24}$ | $73.80_{\pm1.65}$ | $28.53_{\pm1.15}$ | $76.35_{\pm2.18}$ | $80.21_{\pm0.51}$ | $72.23_{\pm2.19}$ | $45.82_{\pm7.52}$ | +26.65 |
| PP | $68.22_{\pm5.68}$ | $72.27_{\pm0.47}$ | $70.15_{\pm5.42}$ | $75.15_{\pm0.83}$ | $79.75_{\pm0.57}$ | $73.83_{\pm8.94}$ | $80.26_{\pm1.45}$ | $78.85_{\pm0.98}$ | $78.51_{\pm0.83}$ | $71.07_{\pm8.49}$ | +2.09 |
| SBAR | $80.99_{\pm1.40}$ | $52.18_{\pm2.15}$ | $80.37_{\pm3.48}$ | $56.32_{\pm6.03}$ | $84.16_{\pm0.56}$ | $40.81_{\pm12.99}$ | $82.17_{\pm0.91}$ | $81.28_{\pm1.06}$ | $82.45_{\pm1.55}$ | $54.46_{\pm4.92}$ | +22.67 |
| ADVP | $91.87_{\pm0.56}$ | $88.38_{\pm0.97}$ | $91.48_{\pm0.61}$ | $89.78_{\pm1.17}$ | $92.22_{\pm1.01}$ | $88.57_{\pm4.53}$ | $92.11_{\pm0.74}$ | $89.67_{\pm0.93}$ | $90.93_{\pm1.59}$ | $88.07_{\pm0.71}$ | +4.48 |
| ADJP | $71.82_{\pm1.43}$ | $63.08_{\pm1.90}$ | $75.18_{\pm2.85}$ | $61.66_{\pm9.97}$ | $78.39_{\pm1.78}$ | $60.40_{\pm8.03}$ | $75.77_{\pm3.74}$ | $75.55_{\pm2.18}$ | $72.90_{\pm4.19}$ | $65.40_{\pm6.60}$ | +7.93 |
| Δ by Model | +12.42 | | +13.05 | | +20.14 | | +3.90 | | +12.76 | | |

Table 7: Recall on six most frequent constituent types. The recall data is calculated over the English test set. Δ by Type indicates the average recall improvement for the constituent type. Δ by Model indicates the average recall improvement for the PCFG variant.

## A.8 PARAPHRASING PROMPTS

We use the below prompts to generate paraphrases from the `gpt-4o-mini-2024-07-18` model. {lang} is a placeholder for languages. For example, we set {lang}="English" when collecting English paraphrases.

- Create grammatical sentences by shuffling the phrases in the below sentence. The generated sentences must be in {lang}. Use the same word as in the original sentence

- Create grammatical sentences by changing the tense in the below sentence. The generated sentences must be in {lang}. Use the same word as in the original sentence.

- Create grammatical sentences by restating the below sentences in passive voice. The generated sentences must be in {lang}. Use the same word as in the original sentence.

- Create grammatical sentences by restating the below sentences in active voice. The generated sentences must be in {lang}. Use the same word as in the original sentence.

- Create grammatical clefting sentences based on the below sentence. The generated sentences must be in {lang}. Use the same word as in the original sentence.

- Create pairs of interrogative and its answers based on the below sentence. The generated sentences must be grammatically correct and be explicit. The sentences must be in {lang}. Use the same word as in the original sentence. The answer to the questions should be a substring of the given sentence.

- Create pairs of confirmatory questions and its answers based on the below sentence. The generated sentences must be grammatically correct and textually diverse. The sentences must be in {lang}. Use the same word as in the original sentence. The answer to the questions should be a substring of the given sentence.

- Create grammatical sentences by performing the topicalization transformation to the below sentence. The sentences must be in {lang}. Use the same word as in the original sentence.

- Create grammatical sentences by performing the heavy NP shift transformation to the below sentence. The sentences must be in {lang}. Use the same word as in the original sentence.

## A.9 EXAMPLES OF THE COLLECTED PARAPHRASES

The below list contains examples of our collected paraphrases for *Such agency ' self-help ' borrowing is unauthorized and expensive , far more expensive than direct Treasury borrowing , said Rep. Fortney Stark -LRB- D. , Calif. -RRB- , the bill 's chief sponsor ..*

- 'Self-help' borrowing by such agency is unauthorized and expensive, far more expensive than direct Treasury borrowing,' said Rep. Fortney Stark -LRB- D., Calif. -RRB-, the bill's chief sponsor.

- Far more expensive than direct Treasury borrowing is such agency ' self-help ' borrowing, unauthorized and expensive, said Rep. Fortney Stark -LRB- D., Calif. -RRB-, the bill 's chief sponsor.

- Yes, he said it is far more expensive than direct Treasury borrowing.

- What is unauthorized and expensive is such agency 'self-help' borrowing, far more expensive than direct Treasury borrowing, according to Rep. Fortney Stark.

- 'Self-help' borrowing by such agency is considered unauthorized and is regarded as expensive, far more expensive than direct Treasury borrowing," said Rep. Fortney Stark -LRB- D., Calif. -RRB-, who is the chief sponsor of the bill.
- According to Rep. Fortney Stark -LRB- D. , Calif. -RRB- , the bill 's chief sponsor , such agency 'self-help' borrowing is unauthorized and far more expensive than direct Treasury borrowing.

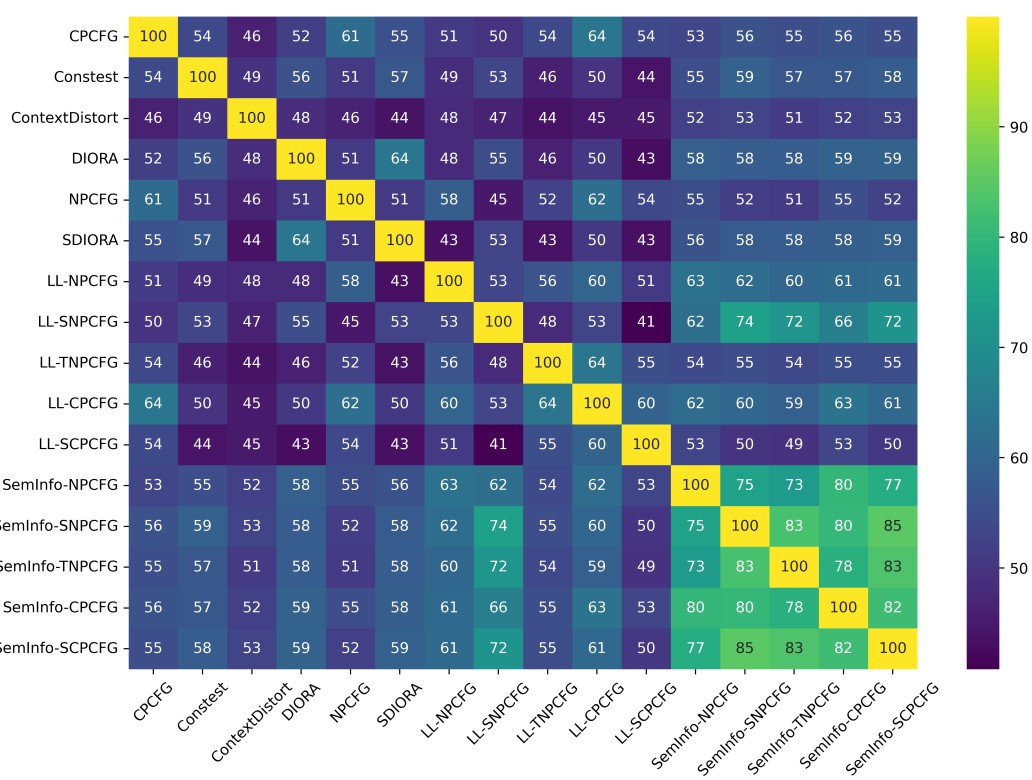

Figure 13: Agreement between heterogeneous parsers

