# OpenReview forum: "Improving Unsupervised Constituency Parsing via Maximizing Semantic Information"
_ICLR.cc/2025/Conference — ICLR 2025 Spotlight_

### Official Review · Reviewer_t1cD · 2024-11-02

**Soundness:** 3
**Presentation:** 4
**Contribution:** 4
**Rating:** 8
**Confidence:** 5

**Summary:**

This work investigates enhancing the performance of Probabilistic Context-Free Grammar (PCFG) parsers in the Unsupervised Constituency Parsing task by incorporating semantic information into the training objective. To achieve this, the authors use the `gpt-4o-mini-2024-07-18` model as an external paraphrasing tool to generate paraphrases for input sentences, assigning semantic scores to individual phrases based on these paraphrases. The parser is then trained with the Reinforce algorithm to maximize the expected total semantic scores of the induced constituents.

This work demonstrates non-ensemble non-distilled state-of-the-art (SOTA) unsupervised parsing performance for English, Chinese, and French, along with substantial performance on German. The authors also conduct additional experiments to show strong correlations between the proposed objective and parsing performance, motivating further investigation of this objective function for non-PCFG models in future work.

**Strengths:**

* Demonstrates the effectiveness of utilizing semantic information in parsing.
* Proposes a novel approach for measuring semantic information through paraphrasing.
* Exhibits significant performance improvements across various setups (different PCFG variants as base models and multiple languages).
* Includes experiments in four languages: English, French, Chinese, and German.
* Focuses on model optimization, making it complementary to other efforts in the field.
* The paper is well-written and easy to follow.

**Weaknesses:**

**I would be happy and am strongly willing to increase my rating if most of the following issues, especially the last one, are fixed.**

* **[Minor Weakness]** The authors frequently use “sentence-F1,” “corpus-level sentence-F1,” and “mean sentence-F1” interchangeably, which may lead to confusion. If I understand correctly, each of these terms refers to the “corpus-level sentence-F1” defined on line 299: “The corpus-level score is computed by averaging the sentence-F1 score across the entire corpus.” In prior research, this metric is commonly referred to as the sentence-level F1 score.

    The commonly accepted distinction, as stated in Footnote 8 of [Kim et al. (2019)](https://aclanthology.org/P19-1228/), is as follows: Corpus-level F1 calculates precision and recall at the corpus level to derive the F1 score, whereas sentence-level F1 calculates F1 for each sentence and then averages across the corpus.

    **Suggestions:** Although this paper defines its terminology, it is recommended to (1) avoid using multiple terms for the same concept, and (2) adopt standard terminology to prevent confusion. If the intent is to differentiate between sentence-level F1 scores used in correlation analysis and the average score used in main experiments, a preferable approach might be to use “sentence-level F1” and “mean sentence-level F1.”

* The paper claims to be “unsupervised” and compares its results with other “unsupervised” models. However, it relies on an external paraphrasing model that may involve either supervised or unsupervised methods of paraphrasing supervision. If this paraphrasing model is supervised, then the entire approach would more accurately be considered an “indirectly” or “distantly supervised” method, which is not directly comparable to fully unsupervised baselines. Specifically, the paraphrasing model used here, `gpt-4o-mini-2024-07-18`, incorporates reinforcement learning from human feedback (RLHF; [OpenAI source](https://openai.com/index/gpt-4o-mini-advancing-cost-efficient-intelligence/)), which makes it a supervised model due to its reliance on human feedback/labels.

    Additionally, one key motivation for claiming an “unsupervised” approach is applicability to low-resource languages or other structure-prediction tasks where GPT models may not be available. This limitation should be more clearly addressed in the paper.

    **Suggestions:** To support the “unsupervised” claim, I suggest conducting experiments with an unsupervised paraphrasing model. A comparable approach is found in [He et al. (2018)](https://arxiv.org/pdf/1808.09111), where their dependency parser model, which requires POS labels, reports two sets of results: one using gold POS tags and the other using induced unsupervised POS tags. This dual-setup approach acknowledges the compounded error introduced by unsupervised POS tagging, providing a clearer comparison.

* Regarding the paraphrasing model, several important questions and concerns should be addressed:
    1. Quality Requirements: What level of performance is required from the paraphrasing model to effectively support the proposed method? Is there a minimum paraphrasing accuracy or quality that significantly impacts the results, and how is this threshold determined?
    2. Robustness to Model Choice: How robust is the proposed model to variations in the paraphrasing model used? For example, would different paraphrasing models (with varying architectures, training datasets, or language capabilities) yield comparable outcomes, or are the results highly sensitive to the choice of the paraphrasing model? This could be demonstrated through experiments with multiple paraphrasing models, both high- and low-performing, to understand the potential variability in results.
    3. Efficiency in Training: How efficient is the training process, given that it requires multiple calls to the paraphrasing model? Discussing any optimizations or alternatives could strengthen the method’s practicality.

    Each of the above points warrants at least some discussion in the paper.

* There is no need for probability notation in Equation (7); the `P` notation can be safely replaced with an indicator function. It is recommended to avoid probability notation here, as `s ∈ MS(x^p, x)` is not a random variable, given that `x^p` serves as the expectation argument.
* In Table 1, the statistical test is conducted using only three samples, which affects its reliability. Since training multiple model instances is resource-intensive, I suggest performing the test at the sentence level. Specifically, consider each sentence’s F1 score as a separate sample, following [Shayegh et al. (2024)](https://aclanthology.org/2024.acl-long.808.pdf).
* The final argument in Section 5.2, which discusses the performance on the German dataset, could benefit from further elaboration. Currently, the claim lacks sufficient support.
* **[Minor Weakness]** In Appendix A.2, the correlation is reported at 10k, 20k, and 30k steps. However, Figure 5 shows that LL-SF1 diminishes significantly after 10k steps, while achieving better correlation at earlier stages. Although this does not affect the conclusion that SemInfo-SF1 maintains a higher correlation throughout the entire training process, it would be beneficial to include results for earlier steps in Appendix A.2 for completeness.
* **[Minor Weakness]** Figure 4 appears to depict a worst-case scenario for LL-SF1 correlation, showing a large negative value. However, Table 2 shows that the correlation is nearly zero for all models and slightly positive on average, which may make Figure 4’s presentation somewhat misleading.
* The paper introduces PCFG models as the state-of-the-art (SOTA) in the task five times, yet only twice (and neither in the Approach nor Introduction sections) specifies that this refers to non-ensemble SOTA. Additionally, the authors claim to achieve new SOTA results four times without clarifying that this pertains only to non-ensemble models.
    1. Transparency of Setup: This approach somewhat obscures the actual setup. It should be clearly stated as non-ensemble, and more importantly, **non-distilled** SOTA, especially as works like [Kim et al. (2019)](https://aclanthology.org/P19-1228/), [Cao et al. (2020)](https://aclanthology.org/2020.emnlp-main.389.pdf), and [Shayegh et al. (2024b)](https://arxiv.org/pdf/2310.01717) demonstrate significant performance improvements by distilling induced knowledge into a URNNG model ([Kim et al.; 2019b](https://arxiv.org/abs/1904.03746)), all of which outperform the best results in this paper. The first two works achieve this by distilling the best model (which requires a validation set), and the latter by distilling ensemble knowledge (non-best, thus without a validation set).
    2. Comparison Justification: It is unclear why the authors chose to compare only with non-ensemble and non-distilled models. This choice is not discussed in the paper. Since distilled models have demonstrated efficiency during inference (Table 5 in [Shayegh et al.; 2024b](https://arxiv.org/pdf/2310.01717)), inference efficiency cannot be the reason. If training efficiency is the concern, then an efficiency analysis on the paraphrasing models is warranted.
    3. Discussion of Related Work: Even if the authors choose not to compare directly with the mentioned works, these studies are significant enough to merit discussion.

    In general, there is an overemphasis on claiming SOTA. This is unnecessary, as the paper’s novelty and valuable contributions stand on their own and are beneficial to the field. Notably, [Shayegh et al. (2024b)](https://arxiv.org/pdf/2310.01717) show that each individual model with specialized expertise can contribute substantially to final parsing performance when part of an ensemble. Specifically, they take advantage from [Li & Lu (2024)](https://scholar.google.com/scholar_url?url=https://arxiv.org/abs/2306.00645&hl=en&sa=T&oi=gsr-r&ct=res&cd=0&d=15034760626359817941&ei=bG4mZ7CxJuG86rQPgN2emQk&scisig=AFWwaeZgwSDS3qGcVtN0rBG9whiK) despite their lower standalone performance, because of its unique approach to Unsupervised Constituency Parsing.

    **Suggestions:**
    * Instead of heavily emphasizing standalone SOTA status, consider including a correlation analysis with other baselines to demonstrate the unique insights offered by the new model, particularly due to the proposed new objective.
    * Incorporating the model into the ensemble approach from [Shayegh et al. (2024b)](https://arxiv.org/pdf/2310.01717) and measuring any resulting performance boost could also provide strong support for the model’s contribution.

**Questions:**

* All the questions in the 3rd point in weaknesses.
* Why is the IDF term included in the SemInfo definition? Is there a specific reason for incorporating this term?
* In Section 5.3.1, why was 30k steps chosen as the evaluation point? Is there a particular rationale behind this choice?

---

> ### Author Response · Authors · 2024-11-23
> **Response to Reviewer t1cD [1/n]**
>
> Thank you for the helpful and detailed feedback. Your suggestions helped improve our manuscript significantly. We have improved our manuscript in accordance with your suggestion and uploaded it to the openreview.
> In the updated version, we added two new experiments: one (Appendix A4) comparing PCFG parsers trained with various paraphrasing models, including one that produces very noisy paraphrases, and the other (Appendix A5) discussing the applicability of our method to the ensemble method.
>
> ---
> > The authors frequently use “sentence-F1,” “corpus-level sentence-F1,” and “mean sentence-F1” interchangeably, which may lead to confusion. If I understand correctly, each of these terms refers to the “corpus-level sentence-F1” defined on line 299: “The corpus-level score is computed by averaging the sentence-F1 score across the entire corpus.” In prior research, this metric is commonly referred to as the sentence-level F1 score.
>
> Sorry for the confusion made. Yes, they are the same. Our motivation was to avoid the confusion of the instance-level sentence-F1 score (which we use for the correlation analysis) with the previously defined corpus-level sentence-F1 metric (which is instance-level sentence-F1 aggregated in the corpus level). In the updated version, we give a unified definition, referring to the corpus-level sentence-F1 (sentence-F1 score defined in previous research) as $\text{SF1}^c$ and the instance-level sentence-F1 $\text{SF1}^i$ in L103.
>
>
>
>
> > The paper claims to be “unsupervised” and compares its results with other “unsupervised” models. However, it relies on an external paraphrasing model that may involve either supervised or unsupervised methods of paraphrasing supervision. If this paraphrasing model is supervised, then the entire approach would more accurately be considered an “indirectly” or “distantly supervised” method, which is not directly comparable to fully unsupervised baselines.
>
> Thank you for pointing this out. Yes, the paraphrasing model may or may not involve paraphrasing supervision. However, the potential paraphrasing supervision is different from constituent parse tree supervision.
> When we discuss whether a model is (weak/distant-)supervised, we are referring to whether it uses labeled data for the task [Wu 2002](https://na.uni-tuebingen.de/ex/ml_seminar_ss2022/Unsupervised_Learning%20Final.pdf). That is, if we were to train a model $f: X\to Y$, we need paired data $(X, Y)$. In the parsing task, $Y$ should be the constituent tree annotation.
> In case of weak-supervised learning, we learn from a collection of labeled data and a set of unlabeled data [Chapelle, 2006](https://www.molgen.mpg.de/3659531/MITPress--SemiSupervised-Learning.pdf). In the case of distant learning, we create pseudo-labeled data from an external database [mintz 09](https://nlp.stanford.edu/pubs/mintz09.pdf).
> However, Our method neither uses nor creates a constituent parse tree for training. This separates us from those (semi/distant)-supervised methods. What we did is to rank all possible parse trees based on their SemInfo value and encourage the PCFG parser to prefer trees with higher SemInfo values. Because of this difference, we weren't able to use the supervised training framework for SemInfo maximization training and had to resort to reinforcement learning instead.
>
>
>
>
> > Additionally, one key motivation for claiming an “unsupervised” approach is applicability to low-resource languages or other structure-prediction tasks where GPT models may not be available. This limitation should be more clearly addressed in the paper.
>
> We agree that our method may not be applicable to all low-resource languages, especially to those where GPT is not available. We added the explanation about GPT's multilingual ability in L190.
>
> However, our claim is that **unsupervised parsers can be improved by maximizing semantic information**, which is orthogonal to the application to low-resource languages.
> The reliance on the GPT model is on our particular estimation of semantic information. If we can find other methods to estimate the semantic information, we can safely remove the GPT model from our pipeline. We will explore GPT-free methods in future studies.
> In addition, the GPT model is a prominent way to perform unsupervised parsing across many languages. The GPT model enjoys an active research community, progressively extending its language support in the past two years. OpenAI's GPT model now officially supports more than 50 languages [openai](https://openai.com/index/gpt-4o-and-more-tools-to-chatgpt-free/), providing boarder coverage than weakly-supervised/distant-supervised methods that rely on existing constituent tree annotations.

---

> > ### Author Response · Authors · 2024-11-23
> > **Response to Reviewer t1cD [2/n]**
> >
> > > Quality Requirements: What level of performance is required from the paraphrasing model to effectively support the proposed method? Is there a minimum paraphrasing accuracy or quality that significantly impacts the results, and how is this threshold determined?
> >
> > We did not find the minimum level of paraphrasing performance needed. In an additional experiment (Appendix A4), we find that our method is robust against paraphrasing noises. We will discuss more details in the next question.
> >
> >
> > >Robustness to Model Choice: How robust is the proposed model to variations in the paraphrasing model used? For example, would different paraphrasing models (with varying architectures, training datasets, or language capabilities) yield comparable outcomes, or are the results highly sensitive to the choice of the paraphrasing model? This could be demonstrated through experiments with multiple paraphrasing models, both high- and low-performing, to understand the potential variability in results.
> >
> > In the additional experiment (Appendix A4), We find that the SemInfo-trained PCFG is robust against paraphrasing noises. We evaluate the parsing accuracy of SemInfo-trained PCFG parsers and MaxTreeDecoding (MTD) parsers using seven paraphrasing models: GPT-4o, gpt-4o-mini, gpt-3.5-turbo (large), llama-3b, qwen-3b (medium), llama-1b, qwen-0.5b models (small). The large, medium, and small groups represent paraphrasing models with different noise levels.
> > We use the MaxTreeDecoding (MTD) method to reflect the paraphrasing quality because its parsing accuracy depends totally on the paraphrasing quality.
> >
> > We observed the following in the experiment:
> > 1. the accuracy degradations of SemInfo-trained PCFG parsers are far less than that of MTD parsers when combined with different paraphrasing models. This indicates our method is robust against noisy paraphrases.
> > 2. the SemInfo-trained PCFG parser significantly outperforms the LL-trained PCFG by 9 points, even when using the nosiest paraphrasing model (qwen2.5-0.5b). This indicates that SemInfo provides useful syntactic information even when it is estimated with noisy paraphrases.
> >
> > > Efficiency in Training: How efficient is the training process, given that it requires multiple calls to the paraphrasing model? Discussing any optimizations or alternatives could strengthen the method’s practicality.
> >
> > The SemInfo maximization training is around twice as expensive as the LL-maximization training. The reason is the need to compute the span-posterior probability through back-propagation.
> > With the optimized code provided by [liu 2024](https://arxiv.org/abs/2310.14997), the complete training process takes around ten hours in a single V100 GPU or 14 hours in a single P100 GPU.
> > The paraphrase collections are more computationally expensive. Yet, our method can achieve high parsing accuracy at a relatively low cost. With the gpt-4o-mini model, the paraphrase collection for English costs around 5 USD and generates around 80M tokens.
> > The same collection takes around 20 hrs using llama 3b model running on 4xA4000 GPUs (1k tokens/s throughput)
> >
> > > There is no need for probability notation in Equation (7); the `P` notation can be safely replaced with an indicator function. It is recommended to avoid probability notation here, as `s ∈ MS(x^p, x)` is not a random variable, given that `x^p` serves as the expectation argument.
> >
> > Thank you for the suggestion. We agree that (7) is excessive. We have removed it from the equation.

---

> > > ### Author Response · Authors · 2024-11-23
> > > **Response to Reviewer t1cD [3/n]**
> > >
> > > > - In Table 1, the statistical test is conducted using only three samples, which affects its reliability. Since training multiple model instances is resource-intensive, I suggest performing the test at the sentence level. Specifically, consider each sentence’s F1 score as a separate sample, following [Shayegh et al. (2024)](https://aclanthology.org/2024.acl-long.808.pdf).
> > >
> > > Thank you for the thoughtful suggestion. The t-test we conduct to evaluate the statistical significance does take the sample count into consideration. When fewer samples are included in the test, a larger accuracy difference is required to achieve statistical significance.
> > >
> > > We did implement the statistical test using the instance-level sentence-F1 score. In this test, all differences are statistically significant. The t-statistic for the instance-level test is shown in the below table.
> > >
> > > | t-statistic | NPCFG | SNPCFG | TNPCFG | CPCFG | SCPCFG |
> > > | ----------- | ----- | ------ | ------ | ----- | ------ |
> > > | English     | 42.74 | 27.62  | 39.78  | 37.06 | 56.91  |
> > > | Chinese     | 43.41 | 30.98  | 26.04  | -4.12 | 21.83  |
> > > | German      | 21.03 | 39.09  | 18.86  | 17.95 | 11.11  |
> > > | French      | 12.74 | 15.8   | 39.31  | 13.26 | 22.24  |
> > >
> > > However, we do find the instance-level test put much more weight on sentence variations than on model variations (as the degree of freedom in the t-statistics comes mainly from the sentence variations). This means the test could not properly reflect the influence on parsing accuracy from the SemInfo-maximization objective.
> > >
> > >
> > > >- The final argument in Section 5.2, which discusses the performance on the German dataset, could benefit from further elaboration. Currently, the claim lacks sufficient support.
> > >
> > > Thank you for pointing out the issue. The discussion is based on the out-of-vocabulary word ratio in the validation set. The rate in the German dataset is 14\%, while the rate is 5\%, 6\%, and 7\% in the English, Chinese, and French dataset, respectively. We added the data in L353.
> > >
> > >
> > > > - **[Minor Weakness]** In Appendix A.2, the correlation is reported at 10k, 20k, and 30k steps. However, Figure 5 shows that LL-SF1 diminishes significantly after 10k steps, while achieving better correlation at earlier stages. Although this does not affect the conclusion that SemInfo-SF1 maintains a higher correlation throughout the entire training process, it would be beneficial to include results for earlier steps in Appendix A.2 for completeness.
> > >
> > > Sorry for the confusion. The diverging coefficient shown in Figure.6 (originally Figure.5) occurs in the corpus-level correlation analysis. In Appendix A2, we discuss the sentence-level correlation. This difference can be seen in the correlation coefficient. In Figure.6,  the coefficient is still at 0.4 level. In comparison, the coefficient is always close to 0 in Figure.7 (originally Figure.6)
> > >
> > > >- **[Minor Weakness]** Figure 4 appears to depict a worst-case scenario for LL-SF1 correlation, showing a large negative value. However, Table 2 shows that the correlation is nearly zero for all models and slightly positive on average, which may make Figure 4’s presentation somewhat misleading.
> > >
> > > Thank you for pointing out the issue. We added more examples in Figure.7.
> > >
> > > > The paper introduces PCFG models as the state-of-the-art (SOTA) in the task five times, yet only twice (and neither in the Approach nor Introduction sections) specifies that this refers to non-ensemble SOTA. Additionally, the authors claim to achieve new SOTA results four times without clarifying that this pertains only to non-ensemble models.
> > >
> > > Thank you for the suggestion. We have downgraded our claim to achieving SOTA-level performance as a non-ensemble model. We add explanations that the SOTA of PCFG induction only applies to non-ensemble models. We also add a discussion with the ensemble model in the related work section (L464)

---

> > > > ### Author Response · Authors · 2024-11-23
> > > > **Response to Reviewer t1cD [4/n]**
> > > >
> > > > > Transparency of Setup: This approach somewhat obscures the actual setup. It should be clearly stated as non-ensemble, and more importantly, **non-distilled** SOTA, especially as works like [Kim et al. (2019)](https://aclanthology.org/P19-1228/), [Cao et al. (2020)](https://aclanthology.org/2020.emnlp-main.389.pdf), and [Shayegh et al. (2024b)](https://arxiv.org/pdf/2310.01717) demonstrate significant performance improvements by distilling induced knowledge into a URNNG model ([Kim et al.; 2019b](https://arxiv.org/abs/1904.03746)), all of which outperform the best results in this paper. The first two works achieve this by distilling the best model (which requires a validation set), and the latter by distilling ensemble knowledge (non-best, thus without a validation set).
> > > >
> > > > We did not compare with the ensemble and distilling methods mainly because our method is orthogonal to the two methods. We now added clarification that our SOTA result is limited to non-ensemble methods.
> > > >
> > > > Firstly, we argue that our method has the potential to ensemble with other base unsupervised parsers.
> > > > 1. We add a new parsing agreement analysis in Appendix.A5. More details can be found in the updated manuscript.
> > > > 2. We use the same setting as   [Shayegh et al. (2024b)](https://arxiv.org/pdf/2310.01717) for the agreement analysis. In this analysis, we find that the agreement score among SemInfo-trained PCFGs is similar to the agreement of the homogeneous parsers reported by  [Shayegh et al. (2024b)](https://arxiv.org/pdf/2310.01717). Also, we find the agreement scores between the SemInfo-trained PCFGs and other base parsers (e.g., DIORA) are similar to the scores among heterogeneous parsers reported. The similarity suggests that our method can be combined with the ensemble method for better performance.
> > > > 3. We were unable to perform a full ensemble evaluation due to the time constraint (as we need to train all heterogeneous parsers from scratch to perform a full analysis). We will continue the ensemble evaluation after the discussion period.
> > > >
> > > > As for the distillation model, we first argue that our method can be combined with the distillation method. We are preparing the experiment and will add it to the appendix later.
> > > > However, we were hesitant towards the URNNG distillation because we observed a strong right-branching bias during our URNNG training attempts. The right branching bias might be beneficial for obtaining high parsing accuracy in English (due to English being strongly right-branching https://arxiv.org/abs/1909.09428), but not in other languages. This point can be seen from ([Kim et al.; 2019b](https://arxiv.org/abs/1904.03746))'s disappointing result on parsing Chinese with URNNG. The situation might worsen in non-right-branching languages such as German. To the best of our knowledge, URNNG distillation wasn't proved useful beyond the English language.
> > > >
> > > >
> > > > > Comparison Justification: It is unclear why the authors chose to compare only with non-ensemble and non-distilled models. This choice is not discussed in the paper. Since distilled models have demonstrated efficiency during inference (Table 5 in [Shayegh et al.; 2024b](https://arxiv.org/pdf/2310.01717)), inference efficiency cannot be the reason. If training efficiency is the concern, then an efficiency analysis on the paraphrasing models is warranted. Discussion of Related Work: Even if the authors choose not to compare directly with the mentioned works, these studies are significant enough to merit discussion.
> > > >
> > > > Thank you for raising the issue. We have added a discussion with the ensemble method in Section 6. We defer the discussion of the URNNG distillation method due to its limited applicability of URNNG to other languages.
> > > >
> > > >
> > > > > - Why is the IDF term included in the SemInfo definition? Is there a specific reason for incorporating this term?
> > > >
> > > > The IDF term measures the substring-semantics information under the PWI framework. The idea is that a substring will carry little information about the semantics if that substring is frequently generated across paraphrases for different sentences. The inclusion of the IDF term can effectively downweigh non-constituent but frequent substrings such as `is an` in sentence like `A is an B`.
> > > > In our experiment, the IDF term does not provide much information. This is because long substrings are rare in small-size corpora like the PTB. Yet, as the corpus size grows, we expect the IDF term to provide more accurate information on how useful a string is in representing the sentence semantics.
> > > >
> > > >
> > > > > In Section 5.3.1, why was 30k steps chosen as the evaluation point? Is there a particular rationale behind this choice?
> > > >
> > > > In preliminary experiments, we found that 30k is around the point where the +-5% of its maximum performance (i.e., the model converges). Due to the high training/storage cost, we truncate the training process at 30k for the correlation analysis.

---

> > > > > ### Comment · Reviewer_t1cD · 2024-11-28
> > > > >
> > > > > Dear authors,
> > > > >
> > > > > Having another look into the added parts to the appendix, I just found a minor confusing sentence with potential to improve. In L960 you mention
> > > > >
> > > > > > We can observe that the agreement score between our SemInfo-trained PCFG parsers ranges from 54-58,
> > > > >
> > > > > Which is confusing as the agreement score between SemInfo-trained PCFG parsers means the bottom-right of Figure 13 to me. I believe you mean the agreement score of your SemInfo-traned PCFG parsers with/against/with respect to other parsers rages from 54-58. Please let me know if I am mistaking. Otherwise, you may consider a small edition.
> > > > >
> > > > > Best,

---

> > > > > > ### Author Response · Authors · 2024-11-28
> > > > > > **Response to Reviewer t1cD**
> > > > > >
> > > > > > Thank you so much! And sorry for the confusion. We have corrected it in the latest update.
> > > > > >
> > > > > > > I believe you mean the agreement score of your SemInfo-traned PCFG parsers with/against/with respect to other parsers rages from 54-58.
> > > > > >
> > > > > > Yes, you are correct. In the latest update, we changed from
> > > > > > >>We can observe that the agreement score between our SemInfo-trained PCFG parsers ranges from 54-58
> > > > > >
> > > > > > to
> > > > > >
> > > > > > >>We can observe that the agreement score between our SemInfo-trained PCFG parsers and other parsers ranges from 54-58 (top-right corner of Figure 13).

---

> > ### Comment · Reviewer_t1cD · 2024-11-23
> > **Different Interpretations of Weak Supervision**
> >
> > In your description of weak-supervised learning as "learn from a collection of labeled data and a set of unlabeled data," it appears to align more closely with the definition of semi-supervised learning ([Chapelle, 2006](https://www.molgen.mpg.de/3659531/MITPress--SemiSupervised-Learning.pdf)). In contrast, weak supervision is often defined as the generation of pseudolabels through heuristics or transformations from another task using guiding principles. These pseudolabels are then used as a form of supervision.
> >
> > For instance, [Shin et al. (ICLR 2022)](https://openreview.net/pdf?id=YpPiNigTzMT) describe weak supervision as follows:
> >
> > "These weak sources include small pieces of code expressing heuristic principles, crowdworkers, lookups in external knowledge bases, pretrained models, and many more (Karger et al., 2011; Mintz et al., 2009; Gupta & Manning, 2014; Dehghani et al., 2017; Ratner et al., 2018). Given an unlabeled dataset, users construct a set of labeling functions (LFs) based on weak sources and apply them to the data. The estimates produced by each LF are synthesized to produce pseudolabels that can be used to train a downstream model."
> >
> > Similarly, [Ratner et al. (2019)](https://ai.stanford.edu/blog/weak-supervision/) explain weak supervision as involving "weaker forms of supervision, such as heuristically generating training data with external knowledge bases, patterns/rules, or other classifiers." and further note that "weak supervision is about leveraging higher-level and/or noisier input from subject matter experts."
> >
> > In your case, the ranking obtained through SemInfo values (derived from paraphrasing) can be viewed as pseudolabels akin to those described above:
> >     What we did is to rank all possible parse trees based on their SemInfo value and encourage the PCFG parser to prefer trees with higher SemInfo values.
> >
> > That said, it is worth noting that terminologies in this area can be somewhat subjective and context-dependent. For example, some may argue that even traditional unsupervised constituency parsing methods rely on forms of language-modeling supervision, rendering them not strictly "unsupervised."
> >
> > Nevertheless, the robustness you have demonstrated in addressing the paraphrasing model effectively resolves my primary concerns. Thank you for your thoughtful clarification.

---

> ### Comment · Reviewer_t1cD · 2024-11-23
> **Overall Comment on the Revision**
>
> Thank you for your dedicated effort in revising the paper and your attention to ensuring its quality. After reviewing the responses to all the reviews and examining the revisions, I am pleased to increase my evaluation score and offer further support for this work.
>
>     We were unable to perform a full ensemble evaluation due to the time constraint (as we need to train all heterogeneous parsers from scratch to perform a full analysis). We will continue the ensemble evaluation after the discussion period.
> In my opinion, the new Appendix A5 is satisfactory, and there is no need for additional experiments in this regard.
>
> I would also like to commend the authors for their significant efforts to improve the quality of their work, even after receiving very positive initial comments. This demonstrates their commitment to excellence and should be recognized and encouraged.
>
> I look forward to continuing the discussion on (semi-/distant-/weakly-/un-)supervision, irrespective of the paper’s evaluation.
>
> Best regards.

---

> > ### Author Response · Authors · 2024-11-23
> > **Thank you**
> >
> > Thank you very much for the thoughtful and encouraging feedback. We deeply appreciate the time and effort you have invested in reviewing our manuscript. Your comments have been invaluable in refining our work. We are encouraged by your interest in further discussions and look forward to continuing the dialogue in the future.
> >
> > Best regards,

---

> ### Author Response · Authors · 2024-11-23
> **Acknowledgement of Interpretation Differences and Additional Details on Training with Pseudo Labels**
>
> Thank you for the detailed explanation. We are happy to see that the paraphrasing result addresses the primary concern.
>
> Yes, the rank deduced from the SemInfo value serves as the optimization signal. Depending on the perspective, this can be seen as a weak supervision signal. Our preliminary experiments suggest that this rank information provides better performance boost than creating pseudo labels using thresholds.
>
> In the early stage of our research, we experimented with a semi-supervised-ish method. Specifically, instead of building a TreeCRF model and optimizing using REINFORCE, we applied supervised learning to maximize $P(s \text{ is constituent}|x)$ if the SemInfo of $s$ is above certain threshold. This method, inspired by [stern 2017](https://aclanthology.org/P17-1076.pdf), creates pseudo labels by marking all substrings with SemInfo above a threshold as constituents.
>
> However, this method consistently underperformed, scoring approximately 5 points lower than our proposed method (the best PCFG model scores 61 instead of 65-67 as shown in Table.2). In some cases, this method even falls short of the LL-trained baseline. We believe this underperformance can be attributed to the paraphrasing noise, which prompted us to adopt the REINFORCE-based optimization in our final method.
>
> The above is just some stories in the development of our method. We will continue to explore the removal of human involvements from our method in the future. Thank you for all the comments and suggestions!

---

### Official Review · Reviewer_M3gk · 2024-11-03

**Soundness:** 3
**Presentation:** 3
**Contribution:** 3
**Rating:** 6
**Confidence:** 4

**Summary:**

This paper presents a new approach to learning an unsupervised constituency parser.  The approach works by a novel distillation procedure that leverages an LLM-based paraphrasing model.  The connection between paraphrasing and syntax is nicely motivated by linguistic theory.  Their approach combines a nice mix of old (keyword extraction, conditional random fields) and new (LLMs).  They provide strong empirical results in five languages over state-of-the-art methods.

**Strengths:**

This paper presents a new approach to learning an unsupervised constituency parser.  The approach works by a novel distillation procedure that leverages an LLM-based paraphrasing model.  It is a very clever idea!  The connection between paraphrasing and syntax is nicely motivated by linguistic theory.  Their approach combines a nice mix of old (keyword extraction, conditional random fields) and new (LLMs).  They provide strong empirical results in five languages over state-of-the-art methods.

**Weaknesses:**

I found the mathematical exposition imprecise and hard to follow. This paper would benefit significantly from iteration with colleagues to identify and address these issues.  This is my main hesitation about this paper and why I gave it an overall score of 6 instead of 8.

**Questions:**

What is the tree probability (line 152) proportional to the sum of potentials? Normally, it is proportional to the product (or exponentiated sum) of potentials.

Can you say a bit more about how the Tree CRF is parameterized? What are the features and grammar?

How stable are the results of changes in the paraphrasing model?

At the top of page 6, you suggest that adding log Z term to the objective stabilizes during *early* training; you could better support that claim if you dropped the log Z term after pre-training. Have you considered trying this?

Please explain in section 3 that the paraphrasing model is taken as a given.  I was confused about that until I got to the experiments section (page 6).

---

> ### Author Response · Authors · 2024-11-23
> **Response to Reviewer M3gk**
>
> Thank you for the helpful and detailed feedback. We have improved our manuscript in accordance with your suggestion and uploaded it to the openreview. We added Figure.4 in the manuscript showing our training pipeline and added an explanation for the pipeline.
>
> ---
> >I found the mathematical exposition imprecise and hard to follow. This paper would benefit significantly from iteration with colleagues to identify and address these issues.
> >Can you say a bit more about how the Tree CRF is parameterized? What are the features and grammar?
>
> Sorry for the confusion. We have added the pipeline of our model in Figure.4, and improved our explanations in Section.4 (L266). Our TreeCRF model is parameterized based on the underlying PCFG model. Specifically, we define the potentials using the probabilities derived from the PCFG. Below, we list the steps we take when training the TreeCRF model.
> Our method consists of the following steps:
> 1. calculating $log(Z(x))$ by applying the inside algorithm on the PCFG model
> 2. compute $P(s \text{ is a consitituent}|x)$ for all spans by using back-propagation.
> 3. set the substring-potential function $\phi(s)=\exp(P(s \text{ is a consitituent}|x))$
> 4. calculate the constituent tree potential as the product of the substring potentials $\phi(t)=\prod_{s\in t} \phi(s)$
> 5. compute the tree distribution $P(t|x)=\frac{\phi(t)}{\sum_{t^\prime}\phi(t^\prime)}$
> 6. Optimize $\mathcal{J}(D)$ based on samples from the tree distribution and their SemInfo value
>
> > What is the tree probability (line 152) proportional to the sum of potentials? Normally, it is proportional to the product (or exponentiated sum) of potentials.
>
> Thank you for pointing out the mistake. We realize that we incorrectly referred to 'potential functions' instead of 'feature functions.' We have corrected the notation in the updated manuscript.
>
> >How stable are the results of changes in the paraphrasing model?
>
> In an additional experiment (Appendix A4), we found the SemInfo-trained PCFG parsers robust against the change in the paraphrasing model and the paraphrasing quality.
> We evaluate the SemInfo-trained PCFG models using seven paraphrasing models.
> We use the MaxTreeDecoding (MTD) method to reflect the paraphrasing quality because its parsing accuracy depends totally on the paraphrasing quality.
> We make two observations:
> (1) the SemInfo-trained PCFG parser is robust against paraphrasing noises, showing less severe accuracy degradation than the MTD parser.
> (2) the PCFG parser can benefit from SemInfo estimated from noisy paraphrases, outperforming the LL-trained PCFG parser by a large margin when using the most noisy paraphrasing model.
>
> >At the top of page 6, you suggest that adding log Z term to the objective stabilizes during _early_ training; you could better support that claim if you dropped the log Z term after pre-training. Have you considered trying this?
>
> Thank you for this insightful suggestion. We agree that investigating the impact of decreasing the contribution of the $log(Z)$ term after pre-training could provide valuable insights. We are currently implementing this experiment and will include the results in the appendix as soon as they are available.
>
>
> > Please explain in section 3 that the paraphrasing model is taken as a given. I was confused about that until I got to the experiments section (page 6).
>
> Thank you for pointing this out. We have updated Section 3 (Line 187) to clarify that the paraphrasing model is considered given in our framework.

---

> > ### Comment · Reviewer_M3gk · 2024-11-26
> >
> > > We added Figure.4 in the manuscript showing our training pipeline and added an explanation for the pipeline.
> >
> > This is a nice addition!
> >
> > I don't understand why you would want to exponentiate a probability in the $\phi$ definition - why not use the probability directly without the $\exp$?
> >
> > > In an additional experiment (Appendix A4)
> >
> > This is great.

---

> > > ### Author Response · Authors · 2024-11-27
> > > **Response to Reviewer M3gk**
> > >
> > > Thank you for the suggestion. We are glad that the newly added figure eliminates some of the ambiguity.
> > >
> > > > I don't understand why you would want to exponentiate a probability in the $\phi$ definition - why not use the probability directly without the $\exp$?
> > >
> > > Thank you for pointing out the issue. We realized the over-smoothing issue that might be caused by applying $\exp$ to $P(s \text{ is a constituent}|x)$. We checked our code and found the code did not apply $\exp$ to $P(s \text{ is a constituent}|x)$. We have corrected the manuscript and will add comparisons with different parameterizations of $\phi$ later to the appendix.
> > >
> > > In our implementation, we used the log-span-posterior probability instead of the span-posterior probability as the feature function. This means the true equation applied in our implementation is as follows, just as suggested in the comment.
> > >
> > > \begin{align}
> > > P(t|x)&\propto \exp(\sum_{s\in t}\log P(s \text{ is a constituent}|x))=\prod_{s\in t} P(s \text{ is a constituent}|x)
> > > \end{align}

---

> > > > ### Comment · Reviewer_M3gk · 2024-11-27
> > > >
> > > > Great.  That makes more sense.
> > > >
> > > > Next question: Why are you using the product of span probabilities to define $p(t \mid x)$ instead of the conditional distribution $P_{\text{cfg}}(t \mid x)$ defined by the PCFG?

---

> ### Author Response · Authors · 2024-11-27
> **Response to Reviewer M3gk**
>
> Thank you for the question.
> The use of the TreeCRF model is mainly based on ease of implementation.
>
>  >  Why are you using the product of span probabilities to define $p(t|x)$ instead of the conditional distribution $P_{cfg}(t|x)$ defined by the PCFG?
>
> We made the choice mainly because we can easily sample from the TreeCRF distribution and can easily evaluate its entropy using [Kim (2019)](https://aclanthology.org/N19-1114/)'s implementation.
> While the sampling from the PCFG distribution can be done similarly to our method [(Johnson 2007)](https://aclanthology.org/N07-1018.pdf), the same cannot be said for the entropy evaluation. We have yet to figure out an efficient way to compute the entropy of a PCFG distribution.
>
> Another concern is the space size of $P_{cfg}(t|x)$. The PCFG defines a distribution over labeled tree  (e.g., *(NT_1 (NT_2 the man) (NT_3 takes (NT_4 the book)))* ), whereas the TreeCRF defines a distribution over unlabeled tree. The cardinality of the labeled tree space is $O(n^{|NT|})$ times larger than the unlabeled tree space, given a sentence with $n$ words. This is rather problematic for the big PCFG models (with thousands of non-terminals). In our preliminary experiments, we did observe some instabilities in the sampling process, and thus opted for the TreeCRF model.
>
> The marginalization of $P_{cfg}(t|x)$ based on the non-terminal labels is non-trivial too. We have yet to figure out how to perform sampling in the label-marginalized $P_{cfg}(t|x)$ distribution.

---

> ### Comment · Reviewer_M3gk · 2024-11-27
>
> > We made the choice mainly because we can easily sample from the TreeCRF distribution and can easily evaluate its entropy using Kim (2019)'s implementation. While the sampling from the PCFG distribution can be done similarly to our method (Johnson 2007), the same cannot be said for the entropy evaluation. We have yet to figure out an efficient way to compute the entropy of a PCFG distribution.
>
> Computing the entropy $H(t | x)$ is straightforward.  It is explained in Li & Eisner (2009) as well as many other works. That paper has a shortcut similar to the gradient trick that you seem to like.  I am happy to explain it if you like.
>
> > Another concern is the space size of $P_{cfg}(t|x)$. The PCFG defines a distribution over labeled tree (e.g., (NT_1 (NT_2 the man) (NT_3 takes (NT_4 the book))) ), whereas the TreeCRF defines a distribution over unlabeled tree. The cardinality of the labeled tree space is $O(n^{|NT|})$ times larger than the unlabeled tree space, given a sentence with $n$ words. This is rather problematic for the big PCFG models (with thousands of non-terminals). In our preliminary experiments, we did observe some instabilities in the sampling process, and thus opted for the TreeCRF model.
>
> I am confused - why do you need to represent the exponentially large space?
>
> > The marginalization of $P_{cfg}(t|x)$ based on the non-terminal labels is non-trivial too.
> > We have yet to figure out how to perform sampling in the label-marginalized $P_{cfg}(t|x)$ distribution.
>
> To sample an unlabeled tree, you can sample a labeled $t \sim p_{cfg}( . \mid x)$ and drop the labels on it (i.e., "un-label" the tree $t$, :-P).  This un-labeled tree ought to be a more accurate representation of trees from the PCFG than sampling from some other intermediate distribution, as you have.  The marginals of the distribution are equivalent to summing over the nonterminal (i.e., $\mathrm{Pr}_{t \sim pcfg(\cdot \mid x)}((i,k) \in dropLabels(t)) = \frac{1}{Z} \sum_X \beta(i,X,k)$).

---

> > ### Author Response · Authors · 2024-11-28
> > **Response to Reviewer M3gk**
> >
> > Thank you for the suggestions.
> >
> > > It is explained in Li & Eisner (2009) as well as many other works.
> >
> > Thank you for pointing out the reference. The Li & Eisner paper (First- and Second-Order...Translation Forests) does appear to provide a gradient-based method for computing H(t|x). This is a good news for us since we were searching for something similar all along. We are now thinking of adding additional experiments using the PCFG model alone. Thank you!
> >
> > > I am confused - why do you need to represent the exponentially large space?
> >
> > Sorry for the confusion caused. Our concern is more on the thinning of probability when we use the labeled tree space. Since the labeled tree space is $O(n^{|NT|})$ times larger than the unlabeled tree space, the log probability of labeled trees would presumably be $O(|NT|\log n)$ times larger in scale ($\log P_{labeled}(t|x)\propto |NT|\log n \log P_{unlabeled}(t|x)$). This would lead to unmanageable RL loss because the REINFORCE gradient is $\nabla \mathbb{E}_t\left[\log P(t|x)I(t, Sem(x))\right]$. This problem doesn't exist in the unlabeled tree space because of it relatively small space.
> >
> > > To sample an unlabeled tree, you can sample a labeled tree and drop the labels on it
> >
> > Thank you for the suggestion. Yes, the sampling from the unlabeled tree distribution can be approximated by sampling from the labeled distribution and dropping the label (was trapped in the sampling-from-inside chart mindset😅). Following this lead, performing our SemInfo maximization should be achievable using
> > - sampling unlabeled tree from the labeled distribution
> > - computing the label-marginal probability for the unlabeled trees using the inside algorithm.
> >
> > The only remaining problem would be the computation of the entropy of the unlabeled distribution, which we will continue to research on. We will start the experiment for the direct PCFG optimization as outlined above and will add the result to appendix if time permits.

---

### Official Review · Reviewer_PqLp · 2024-11-03

**Soundness:** 3
**Presentation:** 3
**Contribution:** 3
**Rating:** 8
**Confidence:** 4

**Summary:**

The authors propose a novel method for unsupervised constituency parsing. The method is based on semantic information from text statistics (e.g. tf-idf, and bag-of-words). The main contributions are: i) novel metric to measure the information between syntactic structures and semantic, and ii) grammar induction model based on the semantic metric as a learning objective. The method shows a high correlation between the proposed metric and parsing performance, and the proposed model shows competitive performance compared to the state-of-the-art.

**Strengths:**

- Clear description of background knowledge and related work needed to understand the proposed method.
- The authors perform a  comprehensive comparison of the proposed metric with different models and languages.

**Weaknesses:**

- Dependence of the model on synthetic data (e.g. gpt) for paraphrasing.

**Questions:**

Please address the following questions during the rebuttal:

- Does the proposed model produce different parse trees over multiple runs?  Could the uncertainty impact the performance of the model?
- Could the method use similar sentences (e.g. nearest neighbours) instead of synthetic data? or other sources of data (e.g. translations, multi-lingual). Could this change have a strong effect on performance?
- Please speculate on the combination of the proposed semantic metric with contextual embeddings for taking into account paraphrasing. For example: https://aclanthology.org/P15-1030.pdf

**Details Of Ethics Concerns:**

I have no concerns.

---

> ### Author Response · Authors · 2024-11-23
> **Response to Reviewer PqLp**
>
> Thank you for the encouraging and detailed feedback. We have improved our manuscript in accordance with your suggestion and uploaded it to the openreview.
> In the updated version, we added a new experiment (Appendix A4) comparing PCFG parsers trained with various paraphrasing models, including one that produces very noisy paraphrases. We also added an agreement analysis evaluating the parser tree agreements among PCFG parsers obtained from multiple training runs, among PCFG parsers using different paraphrasing models, and between PCFG parsers and other heterogeneous parsers.
>
> ---
> > Dependence of the model on synthetic data (e.g. gpt) for paraphrasing.
>
> Thank you for pointing out the issue. Yes, our method depends on the GPT model for paraphrasing. However, we find our method robust to paraphrasing noise and can still improve over the LL-trained PCFG parsers when using very noisy paraphrases. More details can be found In Appendix A4.
>
>
> > Does the proposed model produce different parse trees over multiple runs? Could the uncertainty impact the performance of the model?
>
> Yes, the SemInfo-trained PCFG models produce different parse trees over multiple independent runs. We added an analysis evaluating the agreement between different parsers in Appendix.A5.
> Figure.11 shows the agreement among independent runs. The agreement scores among independent runs are similar (though a bit higher) to the independent runs of DIORA (another unsupervised parser; results are reported in [Shayegh 2024](https://openreview.net/pdf?id=RR8y0WKrFv). This difference suggests a further accuracy improvement using the ensemble method, which we will continue to explore in the future.
>
> So far, we have not observed any negative impact on the model's performance due to this variability.
>
> > Could the method use similar sentences (e.g., nearest neighbors) instead of synthetic data? or other sources of data (e.g. translations, multi-lingual). Could this change have a strong effect on performance?
>
> Yes, but that will depend on the distance metric we choose. For example, the paraphrases are close in the semantic space and hence can be considered to have a small semantic distance. Unfortunately, the semantic distance is not easily translatable to distances deduced from the embedding space of pre-trained models. This is because neighbor strings in the embedding space may not be semantically valid (subsequently, we cannot define semantic distance for them).
> For example, Given the sentence `John likes to date Mary out on a dinner, especially when he is off work`, semantically invalid strings like `John likes on to date Mary out on to a dinner, especially when when is he is off work and receive his bonus salary` are closer to the original sentence (0.922 similarity by all-mpnet-base-v2) than semantically valid strings like `John loves to date Mary out when he is off work` (0.915). Substring frequencies induced from these semantically invalid strings would likely contribute negatively to the SemInfo estimation.
>
> Therefore, we anticipate that using the nearest neighbor in the embedding space will impact performance. Nonetheless, given our method's robustness to paraphrasing noise (as shown in Appendix A4), we are optimistic about its potential with translational data and plan to explore this in future work.
>
>
> > Please speculate on the combination of the proposed semantic metric with contextual embeddings for taking into account paraphrasing. For example: [https://aclanthology.org/P15-1030.pdf](https://aclanthology.org/P15-1030.pdf)
>
> Thank you for the question. We experimented with various embedding-based semantic metrics, but most ended up failing to yield a meaningful semantic metric.
> The problem is that the contextual embeddings work partly by examining lexical overlaps. If one string has a higher lexical overlap with the original sentence than another string, the former string will get a higher score. This is true regardless of whether the former string is semantically valid or not (if semantically invalid, it should carry little semantic information).
> For example, given a target sentence `John likes to date Mary out for a dinner`, the semantically invalid string `to date Mary out for a` has a higher similarity to the target sentence(0.713 using all-mpnet) than the valid substrings `date Mary out` (0.711 using all-mpnet). Since the former string is semantically invalid, it should carry little semantic information. Our method does that because it would assign a maximal frequency of 0 to the former string.
> Nonetheless, we will continue pursuing the application of contextual embeddings to the semantic metric.

---

> > ### Comment · Reviewer_PqLp · 2024-11-23
> > **Response**
> >
> > Thank you for the response, I dont have any more questions.

---

### Official Review · Reviewer_5TGk · 2024-11-04

**Soundness:** 4
**Presentation:** 4
**Contribution:** 4
**Rating:** 8
**Confidence:** 4

**Summary:**

Unsupervised parsing methods typically involve maximizing sentence log likelihood, which does not correlate well with parsing accuracy. In contrast, this paper uses a paraphrase model to define SemInfo, an alternate objective that involves maximizing a mutual information metric between tree spans and sentence paraphrases. This alternate objective can then be used as a replacement for log likelihood when doing PCFG induction. Experiments show that this objective produces substantially better F1 scores than log likelihood maximization across a wide range of PCFG variants and languages.

**Strengths:**

(1) Overall, the paper is strong, and the proposed method is a fundamental improvement in unsupervised parsing. In addition to providing strong results, it provides insight into what is missing in log likelihood maximization and how one might augment log likelihood with other objectives.

(2) The evaluation is very thorough, with 5 PCFG variants and 4 languages, and the method produces very large gains.

(3) The paper is well-written: it is well-organized, does a good job of motivating and explaining the method, and provides useful analysis.

**Weaknesses:**

No major weaknesses; some minor weaknesses are discussed below. I support the acceptance of this paper regardless of whether the additional experiments mentioned below are run or not.

(1) From what I can tell, the paper builds upon Chen et al. (2024), which is a simpler instantiation of the idea that spans are likely to be constituents if they tend to be regenerated by paraphrase models. While this is acknowledged in the related works section, perhaps it is also worth acknowledging the influence of their paper in Sections 2 and/or 3 (unless this paper was concurrent).

(2) I would be curious about ablations for each of the additional components of the method (naive substrings -> maximal substrings, the average baseline and entropy regularization in REINFORCE, the addition of the LL term in SemInfo), for both SemInfo as a training objective and when used for MaxTreeDecoding.

(3) While the paper is motivated as an alternate objective for unsupervised parsing, it instead sort of feels like a method for distilling the grammatical knowledge of a large model (in this case, gpt-4o-mini) into a PCFG. From that perspective, the experiments are not that surprising given that we know that large models have mastery of grammar. The promise of the motivation would be better met if the method used a weaker paraphrase method (e.g., something hand-coded) that does not require already "knowing" the grammar of the language.

**Questions:**

Potential typo: Equation 13: t \in P(t | x) -> t ~ P(t | x)

---

> ### Author Response · Authors · 2024-11-23
> **Response to Reviewer 5TGk**
>
> Thank you for the encouraging and detailed feedback. We have improved our manuscript in accordance with your suggestion and uploaded it to the openreview. In the updated version, we added a new experiment (Appendix A4) comparing PCFG parsers trained with various paraphrasing models, including one that produces very noisy paraphrases.
>
> ----
> > From what I can tell, the paper builds upon Chen et al. (2024), which is a simpler instantiation of the idea that spans are likely to be constituents if they tend to be regenerated by paraphrase models. While this is acknowledged in the related works section, perhaps it is also worth acknowledging the influence of their paper in Sections 2 and/or 3 (unless this paper was concurrent).
>
> Thank you for bringing this to our attention. We agree that Chen et al. (2024) are highly relevant to our work. We have moved the discussion of their paper to the background (L129).
>
>
> >I would be curious about ablations for each of the additional components of the method (naive substrings -> maximal substrings, the average baseline and entropy regularization in REINFORCE, the addition of the LL term in SemInfo), for both SemInfo as a training objective and when used for MaxTreeDecoding.
>
>
> In our preliminary experiment, using naive frequency leads to 0.4 and 1.05 points degradation in the corpus-level SF1 score (see the table below; the corpus-level SF1 score is defined in L103 in the updated manuscript). The small degradation is because the tree maximizing the naive frequency is exactly the tree maximizing the maximal frequency. The identical optimum provides similar optimization targets for the PCFG parser. However, the optimization landscape differs in non-optimum regions. We believe this difference causes the observed accuracy degradation, suggesting that maximal frequency is a more appropriate tool for estimating the SemInfo value.
>
> |                 | NPCFG      | SNPCFG     |
> | --------------- | ---------- | ---------- |
> | Max Frequency   | 64.45±1.13 | 67.15±0.62 |
> | Naive Frequency | 64.07±0.68 | 66.09±0.45 |
>
> In our preliminary experiments, both the baseline and entropy regularization are necessary for successful training. Without the baseline, the PCFG parser would not be trained successfully because of the high variance in the SemInfo value (reward). Without the entropy regularization, the PCFG parser will collapse to a trivial right-branching parser. The ablation of both techniques will result in a parser with an accuracy of ~40 corpus-level SF1 score.
>
> We also find the LL term significant in training the PCFG parser in our preliminary experiment. Our algorithm would likely not produce a meaningful parser without the LL term added in the early training stage.
>
>
> > The promise of the motivation would be better met if the method used a weaker paraphrase method (e.g., something hand-coded) that does not require already "knowing" the grammar of the language.
>
> Thank you for the suggestion. In Appendix A4, we added a comparison of paraphrasing models producing paraphrases of different qualities. We observed that our method is quite robust to paraphrasing noise. Even when using the most noisy paraphrases, our method significantly improves compared to the LL-trained PCFG parser.
> However, we argue that a model with some grammatical knowledge is necessary to predict constituent structures using semantic information. This is because semantic information can be represented in many ways, of which natural language is one that humans use. If we represent semantics in other ways, we may end up with a representation that is different from natural languages ([Chaabouni 2019](https://arxiv.org/pdf/1905.12561))
>
> > Potential typo: Equation 13: t \in P(t | x) -> t ~ P(t | x)
>
> Thank you for pointing out the issue. We have corrected the error.

---

> > ### Comment · Reviewer_5TGk · 2024-11-23
> >
> > Thanks for the detailed response!

---

### Meta-Review · Area_Chair_mhue · 2024-12-22

**Metareview:**

This paper presents a novel method for learning constituency parsers in an unsupervised fashion.  The reviewers are overwhelmingly supportive of acceptance as the consensus is that the paper presents a substantial improvement in this area beyond what exists in the literature.  There are no explicit large weaknesses given the discussion.

**Additional Comments On Reviewer Discussion:**

There is robust discussion between the authors and reviewers for this paper.

---

### Decision · Program_Chairs · 2025-01-22

Accept (Spotlight)